# SILENCER: From Discovery to Mitigation of Self-Bias in LLM-as-Benchmark-Generator

**Peiwen Yuan**[1], **Yiwei Li**[1], **Shaoxiong Feng**[2†], **Xinglin Wang**[1], **Yueqi Zhang**[1],
**Jiayi Shi**[1], **Chuyi Tan**[1], **Boyuan Pan**[2], **Yao Hu**[2], **Kan Li**[1†]

[1] School of Computer Science, Beijing Institute of Technology
[2] Xiaohongshu Inc

{peiwenyuan,liyiwei,wangxinglin,zhangyq,shijiayi,tanchuyi,likan}@bit.edu.cn
{shaoxiongfeng2023}@gmail.com {panboyuan,xiahou}@xiaohongshu.com

## Abstract

LLM-as-Benchmark-Generator methods have been widely studied as a supplement to human annotators for scalable evaluation, while the potential biases within this paradigm remain underexplored. In this work, we systematically define and validate the phenomenon of inflated performance in models evaluated on their self-generated benchmarks, referred to as self-bias, and attribute it to sub-biases arising from question domain, language style, and wrong labels. On this basis, we propose SILENCER, a general framework that leverages the heterogeneity between multiple generators at both the sample and benchmark levels to neutralize bias and generate high-quality, self-bias-silenced benchmark. Experimental results across various settings demonstrate that SILENCER can suppress self-bias to near zero, significantly improve evaluation effectiveness of the generated benchmark (with an average improvement from 0.655 to 0.833 in Pearson correlation with high-quality human-annotated benchmark), while also exhibiting strong generalizability.

## 1 Introduction

The rapid evolution of large language models (LLMs) (Guo et al., 2025; OpenAI, 2025) has led to three key trends in evaluation: (1) benchmarks saturate more quickly, increasing the demand for continual benchmark construction (Wang et al., 2024); (2) LLMs are applied to a wider range of downstream tasks, amplifying the need for more customized benchmarks (Chang et al., 2024); (3) as LLMs achieve and even surpass human capability, ever greater intelligence is demanded of benchmark generators (Patel et al., 2025). Traditional human-centered benchmark construction paradigm struggles to meet these demands due to high annotation costs and the inherent ceiling of human intelligence. To address this, LLM-as-Benchmark-Generator has been proposed as a potential paradigm (Huang et al., 2024; Farchi et al., 2024; Maheshwari et al., 2024; Yuan et al., 2025) for automatically constructing customizable benchmarks that are scalable in difficulty based on LLMs.

However, the potential biases in LLM-generated benchmarks remain underexplored, which may lead to unreliable evaluation results. In this work, we investigate the *self-bias of LLM-as-Benchmark-Generator*, a phenomenon where *LLMs tend to achieve overestimated performance on benchmarks they generate themselves*. Specifically, we first introduce a formal and quantifiable definition of self-bias $\mathcal{B}$, and then empirically demonstrate its widespread presence across various LLMs and evaluation tasks. To gain a deeper understanding of the origin of self-bias, we heuristically disentangle $\mathcal{B}$ into three contributors: **q**uestion domain bias $\mathcal{B}^q$, language **s**tyle bias $\mathcal{B}^s$, and wrong **l**abel bias $\mathcal{B}^l$. Investigative experiments show that the three sub-biases collectively contribute to self-bias, with their relative contributions ordered as $\mathcal{B}^l > \mathcal{B}^q > \mathcal{B}^s$.

---

[†] Corresponding author.

39th Conference on Neural Information Processing Systems (NeurIPS 2025).

To eliminate self-bias for achieving more accurate evaluation results, we propose **SILENCER**, a general **S**elf-b**i**as neutra**l**ization fram**e**work for be**nch**mark gen**er**ation that exploits the heterogeneity of multiple generators to construct self-bias-silenced benchmark. Specifically, to mitigate the sub-biases at sample level, we introduce (1) **Attribute Integration** to encourage the LLM generators to create more domain-unbiased question sets ($\mathcal{B}^q$); (2) **Cross Paraphrase** to avoid stylistic monotony in language ($\mathcal{B}^s$); (3) **Label Calibration** to reduce the inherent alignment between wrong labels and predictions offered by generators ($\mathcal{B}^l$). To suppress self-bias at benchmark level, we propose a **Bias-Neutralizing Ensemble** algorithm, which iterates through two steps: (1) estimating each generator's self-bias by measuring the consistency of its evaluation results with those of the ensembled benchmark, and (2) ensembling the generated benchmarks with weights that are inversely proportional to the estimated self-bias, thereby reducing the self-bias introduced to the final ensembled benchmark.

We validate the effectiveness and generalizability of SILENCER across three tasks, seven LLM generators, and two LLM-as-Benchmark-Generator methods. Comprehensive experimental results demonstrate that SILENCER significantly suppresses self-bias, improving evaluation consistency of LLM-generated benchmark with high-quality human-annotated benchmark by 27.2% (from 0.655 to 0.833) on average. Detailed ablation studies and visualizations confirm the effectiveness and working mechanism of each component in SILENCER. Building on this, we further explore the impact of factors such as the number of generators and benchmark size on self-bias and evaluation effectiveness, offering practical insights for applying SILENCER in real scenarios.

## 2 Related Works

### 2.1 LLM-as-Benchmark-Generator

Benchmark saturation (Glazer et al., 2024), data contamination (Balloccu et al., 2024), and accelerated LLM evolution have collectively motivated the research community to leverage the LLMs for automatic benchmark construction, moving toward the vision of self-evaluation (Maheshwari et al., 2024). In particular, a specific LLM adopts a designated generation strategy to construct a benchmark based on benchmark definition inputs such as the task description and seed samples (Lei et al., 2023; Huang et al., 2024). The evaluation effectiveness of the generated benchmark is typically validated via benchmark agreement testing (Perlitz et al., 2024), where the performance of some LLMs on LLM-generated and human-annotated benchmarks are calculated using certain agreement metric (e.g., rank correlation). Metrics such as sample correctness and diversity are computed to support a more fine-grained evaluation of benchmark quality (Yuan et al., 2025). However, existing studies lack consideration of the generator's self-bias, raising the risk of performance overestimation.

### 2.2 Self-Bias of LLMs

Variations in training corpora, model architectures, and training strategies cause LLMs to exhibit different domain-specific biases (Gallegos et al., 2023). Among them, self-bias generally refers to the tendency of the LLM to favour its own generations when acting as a judge (Panickssery et al., 2024; Xu et al., 2024; Bai et al., 2023). In this work, we extend the notion of self-bias to the setting of LLM-as-Benchmark-Generator. While judge-side self-bias primarily stems from content-based self-preference (Gu et al., 2024), our experiments show that self-bias of benchmark generators arises from multiple, more intricate factors and thus requires mitigation efforts from multiple aspects [*].

## 3 Preliminary

### 3.1 Formal Definition and Verification of Self-Bias

Humans tend to perform better on tests they create themselves (Corrigan and Craciun, 2013; Saucier et al., 2024). This motivates the necessity to investigate whether LLMs show similar self-bias when evaluated on benchmarks they generate, which previous studies have largely neglected (Huang et al., 2024; Yuan et al., 2025). We define $R(\mathcal{M}|\mathcal{M}^{que}, \mathcal{M}^{lab})$ as the performance (e.g. accuracy) of model $\mathcal{M}$ evaluated on benchmarks where questions are provided by $\mathcal{M}^{que}$ and labels are provided by

_________________

[*]Unless otherwise noted, all subsequent mentions self-bias refers to that in LLM-as-Benchmark-Generator.

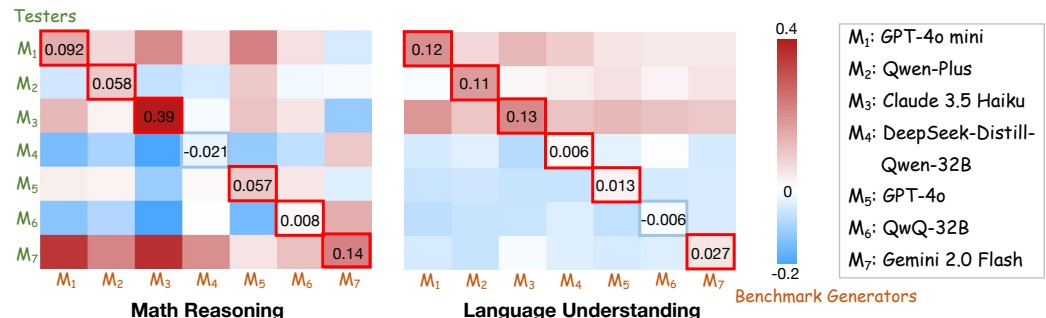

Figure 1: Evaluation bias of LLMs across tasks. The value at position $(\mathcal{M}_i, \mathcal{M}_j)$ represents $\mathcal{B}(\mathcal{M}_j|\mathcal{M}_i)$ (Eq. (2)), the evaluation bias of $\mathcal{M}_j$ on the benchmark generated by $\mathcal{M}_i$.

$\mathcal{M}^{lab}$. As different benchmarks vary in difficulty, to compare the evaluation results across them, we calculate the relative performance of $\mathcal{M}$ with respect to $K$ reference models $\mathcal{M}^{\text{ref}}_{1:K}$ as follows:

$$R\left(\mathcal{M} \mid \mathcal{M}^{\text{que}}, \mathcal{M}^{\text{lab}}, \mathcal{M}^{\text{ref}}_{1:K}\right) = \frac{K \cdot R\left(\mathcal{M} \mid \mathcal{M}^{\text{que}}, \mathcal{M}^{\text{lab}}\right)}{\sum_{k=1}^{K} R\left(\mathcal{M}^{\text{ref}}_k \mid \mathcal{M}^{\text{que}}, \mathcal{M}^{\text{lab}}\right)} \quad (1)$$

On this basis, the evaluation bias of $\mathcal{M}$ on benchmark generated by $\mathcal{M}^{\text{gen}}$ can be calculated as:

$$\mathcal{B}(\mathcal{M}|\mathcal{M}^{\text{gen}}) = R(\mathcal{M}|\mathcal{M}^{\text{gen}}, \mathcal{M}^{\text{gen}}, \mathcal{M}^{\text{ref}}_{1:K}) - R(\mathcal{M}|\mathcal{M}^{\text{human}}, \mathcal{M}^{\text{human}}, \mathcal{M}^{\text{ref}}_{1:K}) \quad (2)$$

where $R(\mathcal{M}|\mathcal{M}^{\text{human}}, \mathcal{M}^{\text{human}}, \mathcal{M}^{\text{ref}}_{1:K})$ denotes the relative performance on human-annotated benchmark. Accordingly, the self-bias of $\mathcal{M}$ can be denoted as $\mathcal{B}(\mathcal{M}|\mathcal{M})$, abbr. $\mathcal{B}(\mathcal{M})$. A larger value of $\mathcal{B}(\mathcal{M})$ indicates a greater degree of self-bias in the benchmark generated by the $\mathcal{M}$.

Based on the definition above, we validate the existence of self-bias on math reasoning and language understanding tasks. We select human-annotated MATH (Hendrycks et al., 2021) and MMLU-Pro (Wang et al., 2024) benchmarks for comparison and employ BenchMaker method (Yuan et al., 2025) for benchmark generation, which has been validated to construct high-quality benchmarks (See a detailed introduction of BenchMaker in Appendix A and exact experimental settings in §5.1). The seven LLMs illustrated in Figure 1 are employed as both generators, testers and reference models, with the results of evaluation bias presented accordingly. Based on Figure 1, we identify two primary findings: (1) Models generally perform better on benchmarks they generate themselves (highlighted by red boxes on the diagonal), exhibiting average self-biases of 0.103 and 0.058 across the two tasks, respectively. (2) Some models demonstrate a prevalent negative evaluation bias on model-generated benchmarks, as reflected by the blue rows (e.g., $\mathcal{M}_4$ in both tasks). We verify (see Appendix B for details) that the reason lies in data contamination issue (Balloccu et al., 2024), which lead to the inflated relative performance $R(\mathcal{M}|\mathcal{M}^{\text{human}}, \mathcal{M}^{\text{human}}, \mathcal{M}^{\text{ref}}_{1:K})$ of these models on human-annotated benchmarks, subsequently resulting in negative evaluation bias according to Eq. (2) [†].

### 3.2 Disentangling of Self-Bias

After confirming the widespread presence of self-bias, we proceed to investigate its underlying causes. Since a benchmark typically consists of a question set and a label set, and its presentation can be influenced by language style, we heuristically decouple self-bias into three sub-biases:

**Language style bias $\mathcal{B}^s$.** *The language style of questions generated by a model may align with what it is familiar with, enabling it to better understand the questions and answer them correctly.* We formally measure $\mathcal{B}^s$ as follows:

$$\mathcal{B}^s(\mathcal{M}) = R(\mathcal{M}|\text{para}_{\mathcal{M}}(\mathcal{M}^{\text{human}}), \mathcal{M}^{\text{human}}, \mathcal{M}^{\text{ref}}_{1:K}) - R(\mathcal{M}|\mathcal{M}^{\text{human}}, \mathcal{M}^{\text{human}}, \mathcal{M}^{\text{ref}}_{1:K}) \quad (3)$$

where we measure how much $\mathcal{M}$ is overestimated on a human-constructed benchmark whose question set has been paraphrased by $\mathcal{M}$.

---

[†]Models without data contamination consequently exhibit underestimated relative performance, as reflected by the red rows (e.g., $\mathcal{M}_1$ in both tasks).

**Question domain bias $\mathcal{B}^q$.** *The domain of questions generated by a model may largely align with the domain it excels in, making it likely to answer them correctly.* We formally measure $\mathcal{B}^q$ as follows:

$$\mathcal{B}^q(\mathcal{M}) = R(\mathcal{M}|\text{para}_{\mathcal{M}^{\text{human}}}(\mathcal{M}), \mathcal{M}^{\text{human}}, \mathcal{M}^{\text{ref}}_{1:K}) - R(\mathcal{M}|\mathcal{M}^{\text{human}}, \mathcal{M}^{\text{human}}, \mathcal{M}^{\text{ref}}_{1:K}) \quad (4)$$

where we measure how much $\mathcal{M}$ is overestimated on a benchmark built from its self-generated questions (while paraphrased by humans to disentangle the impact of $\mathcal{B}^s$) and labels annotated by humans. Given the impracticality of manual annotating across multiple experimental settings, we adopt the powerful OpenAI o3-mini (OpenAI, 2025) as a proxy for human annotators.

**Wrong label bias $\mathcal{B}^l$.** *The model may produce incorrect labels, and the model's tendency to repeat the same mistakes during labeling and answering can lead to many false positives, inflating the accuracy.* We formally measure $\mathcal{B}^l$ as follows:

$$\mathcal{B}^l(\mathcal{M}) = R(\mathcal{M}|\mathcal{M}^{\text{human}}, \mathcal{M}, \mathcal{M}^{\text{ref}}_{1:K}) - R(\mathcal{M}|\mathcal{M}^{\text{human}}, \mathcal{M}^{\text{human}}, \mathcal{M}^{\text{ref}}_{1:K}) \quad (5)$$

where we measure how much $\mathcal{M}$ is overestimated on a benchmark constructed from human-written questions and $\mathcal{M}$-generated labels. In Appendix C.1, we further theoretically prove that models tend to perform better on samples labeled by themselves.

Table 1: Quantitative results of the decoupled sub-biases of $\mathcal{B}$ across tasks.

| | **MATH REASONING** | | | | **LANGUAGE UNDERSTANDING** | | | |
| --- | --- | --- | --- | --- | --- | --- | --- | --- |
| | $\mathcal{B}$ | $\mathcal{B}^s$ | $\mathcal{B}^q$ | $\mathcal{B}^l$ | $\mathcal{B}$ | $\mathcal{B}^s$ | $\mathcal{B}^q$ | $\mathcal{B}^l$ |
| VALUE | 0.1032 | 0.0205 | 0.0247 | 0.0921 | 0.0575 | 0.0061 | 0.0077 | 0.0545 |
| RELATIVE CONTRIBUTION (%) | - | 14.96 | 17.99 | 67.05 | - | 8.967 | 11.29 | 79.74 |

Following the experimental setup as in §3.1, Table 1 reports the magnitudes and relative contributions of each sub-bias. As we hypothesized, all three sub-biases are indeed present and collectively constitute the self-bias illustrated in Figure 1. We further observe that their relative contributions follow the order: $\mathcal{B}^l > \mathcal{B}^q > \mathcal{B}^s$. This indicates that, although both language style and question domain biases have an impact, the high alignment between the model acting as labeler and tester contributes more significantly to the self-bias.

## 4 SILENCER Framework

### 4.1 Motivations

Since different LLMs generally exhibit distinct biases (Gallegos et al., 2023), the most straightforward strategy to mitigate self-bias is to ensemble benchmarks generated by multiple LLMs $\mathcal{M}_{1:T}$ to achieve bias neutralization. However, this method suffers from two limitations. First, each sample still carries all three types of sub-biases from its generator, and the accumulation of these sub-biases leads to a distribution shift away from the unbiased sample domain. Meanwhile, since different LLMs exhibit varying degrees of self-bias, uniformly ensembling their generated benchmarks can be suboptimal. To this end, we propose SILENCER (as shown in Figure 2), a general framework for LLM-as-Benchmark-Generator methods that (1) individually mitigates the three sub-biases at the sample level (§4.2), and (2) performs weighted ensemble of benchmarks from different generators at the benchmark level (§4.3). The prompts used can be found in Appendix E.

### 4.2 Sample-wise Sub-biases Mitigation

**Attribute Integration.** Current LLM-as-Benchmark-Generator methods (Huang et al., 2024; Yuan et al., 2025) generally adopt an attribute-guided approach to ensure the diversity of the question domain (Yu et al., 2023). Specifically, certain generator LLM $\mathcal{M}_t$ is first prompted to generate a set of attributes $A_t$ (e.g., topic) based on the task definition TD, with each attribute corresponding to multiple candidate values (e.g., sports). Afterwards, a list of (attribute, value) pairs is sampled for guiding the LLM to generate a question $q_t$ meeting the attributes, and corresponding label $l_t$:

$$A_t = \text{AttributeGeneration}_{\mathcal{M}_t}(\text{TD})$$
$$(q_t, l_t) = \text{SampleGeneration}_{\mathcal{M}_t}(\text{AttributeSampling}(A_t)) \quad (6)$$

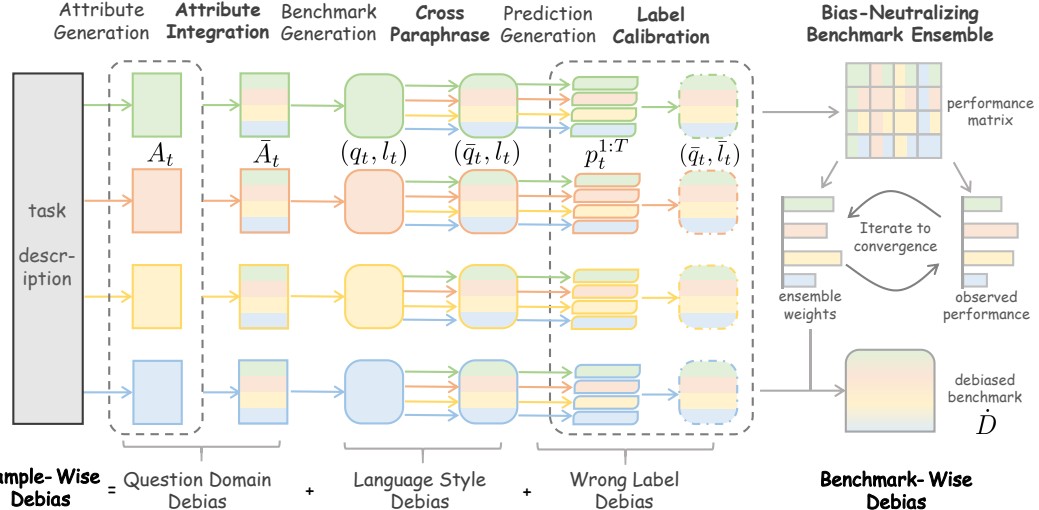

Figure 2: Overall illustration of our SILENCER framework. Each generator LLM is represented by a unique color. Colored arrows indicate the process by which a certain LLM takes the content on the left as input and produces the content on the right as output (excluding the gray arrows).

Since the attributes largely determine the content of the generated question, we look to mitigate the question domain bias of $\mathcal{M}_t$ by suppressing the bias inherently contained in $A_t$:

$$\bar{A}_t = \text{AttributeIntegration}_{\mathcal{M}_t}(A_{1:T}, \text{TD}) \tag{7}$$

where $\mathcal{M}_t$ takes as input the attribute sets produced by all generators $\mathcal{M}_{1:T}$ and integrates the high-quality attributes aligned with the task definition TD, thereby deriving an unbiased attribute set $\bar{A}_t$ for subsequent sample generation.

**Cross Paraphrase.** After each $\mathcal{M}_t$ completes its own benchmark generation $D_t$ based on $\bar{A}_t$, we enhance the linguistic diversity of these benchmarks by having the generators paraphrase each other's question sets without altering the original meaning:

$$\bar{q}_t = \text{QuestionParaphrase}_{\mathcal{M}_r}(q_t), \quad r \in \{1, ..., T\} \text{ chosen uniformly at random} \tag{8}$$

Building on this, the question set of $D_t$ no longer exhibits a single language style, thereby significantly suppressing $\mathcal{B}^s$.

**Label Calibration.** Due to the limitations and invariance in knowledge and capabilities, the label $l_t$ provided by model $\mathcal{M}_t$ may be incorrect, and its predictions tend to exhibit consistent errors, leading to obvious $\mathcal{B}^l$ as shown in Table 1. To this end, we consider leveraging the predictions from other generators $p_t^{1:T}$ to assist $\mathcal{M}_t$ in calibrating the old label $l_t$:

$$\begin{aligned} p_t^i &= \text{Prediction}_{\mathcal{M}_i}(\bar{q}_t) \\ \bar{l}_t &= \text{LabelCalibration}_{\mathcal{M}_t}(\bar{q}_t, l_t, p_t^{1:T}) \end{aligned} \tag{9}$$

Through jointly analyzing and comparing multiple predictions (with their rationales), $\mathcal{M}_t$ is able to produce a more reliable label $\bar{l}_t$, which, even when incorrect, is less likely to align with its own erroneous predictions.

### 4.3 Benchmark-wise Self-bias Suppression

After obtaining the benchmarks $\bar{D}_{1:T}$ in which sub-biases have been mitigated at the sample level, we consider how to ensemble them to further eliminate self-bias. Ideally, benchmarks with less self-bias should be assigned greater weight during ensembling, so that the resulting benchmark exhibits minimal overall self-bias. However, in real-world scenarios, we often lack high-quality human-annotated benchmark $D^{\text{human}}$ to evaluate the self-bias $\mathcal{B}_{1:T}$ of $\bar{D}_{1:T}$. Therefore, we design the Bias-Neutralizing Ensemble Algorithm 1, which uses the ensembled benchmark as a proxy for $D^{\text{human}}$ and iteratively optimizes the weights $\alpha_{1:T}$ used for benchmark ensembling.

---

**Algorithm 1** Bias-Neutralizing Ensemble Algorithm.

---

**Require:** Generated benchmarks $\bar{D}_{1:T}$, Relative performance of $\mathcal{M}_i$ on $\bar{D}_j$ $x_i^j$ (for $i \in [1,T]$, $j \in [1,T]$),
    Desired benchmark size $N$, Minimum value $\epsilon$ (default: 1e-6)
**Ensure:** Ensembled benchmark with low self-bias: $\dot{D}$
 1: $\alpha_i \leftarrow 0$, $\alpha_i^{\text{new}} \leftarrow \frac{1}{T}$ for $i \in [1,T]$
 2: **while** $\sum_{i=1}^{T} |\alpha_i^{\text{new}} - \alpha_i| > \epsilon$ **do**
 3:     $\alpha_{1:T} \leftarrow \alpha_{1:T}^{\text{new}}$
 4:     $\bar{x}_i \leftarrow \sum_{j=1}^{T} \alpha_j \cdot x_i^j$ for $i \in [1,T]$
 5:     $\alpha_{1:T}^{\text{new}} \leftarrow \texttt{UpdateAlpha}(x_{1:T}^{1:T}, \bar{x}_{1:T})$
 6:     $\alpha_i^{\text{new}} \leftarrow \frac{\alpha_i^{\text{new}}}{\sum_{j=1}^{T} \alpha_j^{\text{new}}}$ for $i \in [1,T]$
 7: **end while**
 8: $\dot{D} \leftarrow \bigcup_{i=1}^{T} \texttt{RandomSampling}(\bar{D}_i, \lfloor N \cdot \alpha_i \rfloor)$

---

Specifically, we first initialize the weight $\alpha_i$ as $\frac{1}{T}$. Then, we compute the weighted performance $\bar{x}_i$ of each $\mathcal{M}_i$ on all the generated benchmarks using $\alpha_{1:T}$ (model performance $x_{1:T}^{1:T}$ can be calculated from the predictions in Eq. (9)). Here, $\bar{x}_i$ essentially represents the performance of $\mathcal{M}_i$ on the ensembled benchmark $\dot{D}$ of current iteration. Treating $\dot{D}$ as a proxy for $D^{\text{human}}$, we use the algorithm $\texttt{UpdateAlpha}(\cdot)$ to update $\alpha$. This process is repeated until convergence.

Ideally, if we can design an effective $\texttt{UpdateAlpha}(\cdot)$ such that $\alpha_{1:T}$ exhibits a clear negative correlation with $\mathcal{B}_{1:T}$, then each iteration will yield a ensembled benchmark $\dot{D}$ with lower self-bias. This, in turn, means that $\dot{D}$ can serve as a better proxy for $D^{\text{human}}$, thereby facilitating a more accurate estimation of $\alpha_{1:T}$ in the next iteration. To identify the most effective $\texttt{UpdateAlpha}(\cdot)$, we explore the following three strategies in our early-stage experiments:

**Self-Bias.** Naively, we use $\bar{x}_i$ as the basis and compute the reciprocal of each model's estimated self-bias as its ensemble weight:

$$\texttt{UpdateAlpha}^{\text{SelfBias}}(x_{1:T}^{1:T}, \bar{x}_{1:T})_i = \frac{1}{x_i^i - \bar{x}_i} \tag{10}$$

**Accuracy.** The Dunning-Kruger Effect (Kruger and Dunning, 1999) suggests that individuals with lower competence tend to overestimate their abilities. Thus, we directly use the accuracy as weight:

$$\texttt{UpdateAlpha}^{\text{Accuracy}}(x_{1:T}^{1:T}, \bar{x}_{1:T})_i = \bar{x}_i \tag{11}$$

**Evaluation Consistency.** As stronger self-bias tends to result in less reliable evaluation results, we quantify the self-bias of each generated benchmark by assessing its evaluation consistency with the ensembled benchmark, which in turn determines its ensemble weight:

$$\texttt{UpdateAlpha}^{\text{Consistency}}(x_{1:T}^{1:T}, \bar{x}_{1:T})_i = \text{PearsonCorrelation}(x_{1:T}^i, \bar{x}_{1:T}) \tag{12}$$

To verify the effectiveness of the three strategies, we compute the Pearson correlation between the reciprocal of the ensemble weights derived from each strategy and the true self-bias under the settings described in §3.1. The results show: Consistency$(0.861) >$ Self-Bias$(0.530) >$ Accuracy$(0.343)$, indicating that Evaluation Consistency better reflects the degree of self-bias. We hypothesize that the reason lies in the fact that, when there is a discrepancy between $\bar{x}$ and the true model performance, the Evaluation Consistency strategy can fully leverage the original model performance data $x_{1:T}^{1:T}$ compared to the Self-Bias and Accuracy strategies, thus partially mitigating error accumulation during the iterative process. We also evaluate the correlation between the ensemble weights and the effectiveness of each generated benchmark (effectiveness of $\bar{D}_i$ is defined as the Pearson correlation between reference models' performances on $\bar{D}_i$ and on $D^{\text{human}}$): Consistency$(0.965) >$ Self-Bias$(0.581) >$ Accuracy$(0.241)$. This indicates that the ensemble weights derived from the Evaluation Consistency strategy also better reflect the evaluation effectiveness of each $\bar{D}_i$, thereby enabling the ensembled benchmark $\dot{D}$ to more accurately assess model performance. Based on these findings, we adopt $\texttt{UpdateAlpha}^{\text{Consistency}}(\cdot)$ as our default setting. To ensure the non-negativity of the weights and the convergence of the iteration (the proof of convergence is provided in Appendix C.2), we adopt the following form in practice, where $\epsilon$ is set as 1e-6:

$$\texttt{UpdateAlpha}_{\text{Silencer}}^{\text{Consistency}}(x_{1:T}^{1:T}, \bar{x}_{1:T})_i = \text{ReLU}(\text{PearsonCorrelation}(x_{1:T}^i, \bar{x}_{1:T})) + \epsilon \tag{13}$$

# 5 Experiments

In this section, we conduct a comprehensive evaluation of the proposed SILENCER framework. Specifically, we introduce the detailed experimental settings in §5.1. In §5.2, we validate the effectiveness of SILENCER in mitigating self-bias and enhancing evaluation effectiveness, examine its generalizability across generators, tasks, and LLM-as-Benchmark-Generator methods, and perform ablations on its individual components. In §5.3, we further investigate other impact of SILENCER on the generated benchmark with fine-grained metrics. In §5.4, we explore how varied settings influence the self-bias and evaluation effectiveness of the generated benchmarks in practice.

## 5.1 Experimental Settings

**Tasks.** We select three tasks to evaluate the cross-task effectiveness of SILENCER, with each task paired with a high-quality human-annotated benchmark for comparison: math reasoning (MATH (Hendrycks et al., 2021)), language understanding (MMLU-Pro (Wang et al., 2024)), and commonsense reasoning (HellaSwag (Zellers et al., 2019)).

**Models.** As for generators, we select seven representative open- and closed-source LLMs: GPT-4o mini (OpenAI, 2024b), Qwen-Plus (Qwen-Team, 2024), Claude 3.5 Haiku (Anthropic, 2025), DeepSeek-Distill-Qwen-32B (DeepSeek-AI, 2025), GPT-4o (OpenAI, 2024a), QwQ-32B (Qwen-Team, 2025), and Gemini 2.0 Flash (DeepMind, 2025). As for reference models, to ensure the robustness of experimental results, we additionally include nine more LLMs (see Appendix D) except for generators, resulting in a 16-model reference set. For each setting, we report the averaged result of 1,000 seeds, each time randomly selecting $T$ ($T > 2$) generators and $K$ reference models ($K > 7$) from the candidate pool.

**Baselines.** We primarily adopt BenchMaker (Yuan et al., 2025) as the baseline method for benchmark generation and evaluate its performance under the SILENCER framework. We take the task descriptions used in Yuan et al. (2025) as input and have the generators construct benchmarks of a specified size. We additionally include a simple and broadly applicable baseline method AttrPrompt (Yu et al., 2023) to further assess the cross-method generalizability of SILENCER.

**Details.** Since multiple settings are involved, we default the benchmark size $N$ for each sub-task within a task (e.g., MATH-Algebra) to 50. We explore the impact of larger sizes in Section §5.4. To ensure fairness in the comparison, we keep the benchmark size consistent before and after ensembling. The sampling temperature for the LLMs is set to 1. In addition to self-bias, we also compute the Pearson correlation $r_p$ of reference models' performance between model-generated and high-quality human-annotated benchmarks to assess the evaluation effectiveness of the generated benchmarks.

## 5.2 Effectiveness and Generalizability of SILENCER

**Overall Effectiveness and Ablation Studies.** The main experimental results are shown in Table 2. The benchmarks generated using the baseline methods exhibit significant self-bias, consistent with the findings in §3.1. The results in rows 2-5 demonstrate that our proposed sample-level strategies effectively mitigate different sub-biases, thereby reducing self-bias $\mathcal{B}$ and improving evaluation effectiveness $r_p$. Among them, the Label Calibration strategy yields the largest gain by alleviating the wrong label bias $\mathcal{B}^l$, which is consistent with the conclusion in §3.2 that $\mathcal{B}^l$ is the major contributor to self-bias. Building on the mitigation of sub-biases, directly ensembling the model-generated benchmarks can substantially reduce $\mathcal{B}$ and significantly improve $r_p$ (row 6), which is consistent with our hypothesis in Section 4.1. Meanwhile, after applying our proposed reweighted ensemble strategy, $\mathcal{B}$ is reduced to nearly zero and $r_p$ reaches the highest value across all tasks, verifying the effectiveness of Bias-Neutralizing Ensemble Algorithm.

**Generalizability.** Across all settings, we observe that SILENCER consistently yields significant reductions in $\mathcal{B}$ (from 0.113 to 0.009 on average) and improvements in $r_p$ (from 0.655 to 0.833 on average), demonstrating its generalizability across tasks, generators and methods.

## 5.3 A Closer Look at What SILENCER Brings

To gain deeper insights into how SILENCER affects benchmark generation, we analyze it from four perspectives with fine-grained metrics (on Math Reasoning task by default).

Table 2: Main results of the baseline methods with different modules of SILENCER. $\mathcal{B}$ and $r_p$ denote the average self-bias and evaluation effectiveness (Pearson correlation) across multiple settings. All results of $r_p$ are statistically significant ($p$-value $< 0.05$).

| METHODS | MATH REASONING $\mathcal{B}\downarrow$ | $r_p\uparrow$ | LANGUAGE UNDERSTANDING $\mathcal{B}\downarrow$ | $r_p\uparrow$ | COMMONSENSE REASONING $\mathcal{B}\downarrow$ | $r_p\uparrow$ |
|---|---|---|---|---|---|---|
| BASELINE METHOD: **BENCHMAKER**(YUAN ET AL., 2025) | | | | | | |
| BASELINE | 0.1205 | 0.7863 | 0.0802 | 0.6816 | 0.1102 | 0.6479 |
| w. ATTRIBUTE INTEGRATION: $\mathcal{B}^q$ | 0.1179 | 0.7892 | 0.0748 | 0.6907 | 0.1059 | 0.6557 |
| w. CROSS PARAPHRASE: $\mathcal{B}^s$ | 0.1074 | 0.7952 | 0.0759 | 0.6840 | 0.1107 | 0.6511 |
| w. LABEL CALIBRATION: $\mathcal{B}^l$ | 0.1109 | 0.8302 | 0.0583 | 0.7058 | 0.1021 | 0.6582 |
| w. $\mathcal{B}^q$ $\mathcal{B}^s$ $\mathcal{B}^l$ | 0.1002 | 0.8371 | 0.0517 | 0.7145 | 0.0934 | 0.6627 |
| w. $\mathcal{B}^q$ $\mathcal{B}^s$ $\mathcal{B}^l$ w. ENSEMBLE | 0.0218 | 0.9216 | 0.0076 | 0.7953 | 0.0178 | 0.7826 |
| w. SILENCER | **0.0140** | **0.9386** | **0.0011** | **0.8340** | **0.0053** | **0.8149** |
| BASELINE METHOD: **ATTRPROMPT**(YU ET AL., 2023) | | | | | | |
| BASELINE | 0.1427 | 0.6179 | 0.1148 | 0.5843 | 0.1093 | 0.6106 |
| w. SILENCER | **0.0187** | **0.8641** | **0.0125** | **0.7725** | **0.0074** | **0.7762** |

**Sample Semantic Distribution.** The presence of $\mathcal{B}^l$ and $\mathcal{B}^q$ results in significant discrepancies in the semantic distributions of samples generated by different generators, as shown in Figure 3(a). After applying Cross Paraphrase alone (Figure 3(b)) and both Cross Paraphrase and Attribute Integration simultaneously (Figure 3(c)), the separation among benchmarks from different LLMs is visibly reduced, indicating that $\mathcal{B}^l$ and $\mathcal{B}^q$ have been mitigated. We further compute the average Maximum Mean Discrepancy (Gretton et al., 2012) and Wasserstein Distance (Villani et al., 2008) between benchmarks from different generator pairs, both of which measure the divergence between two distributions. After applying SILENCER, both metrics decrease (from 0.1174 to 0.1025 and from 12.267 to 10.903, respectively), providing additional evidence for this mitigation.

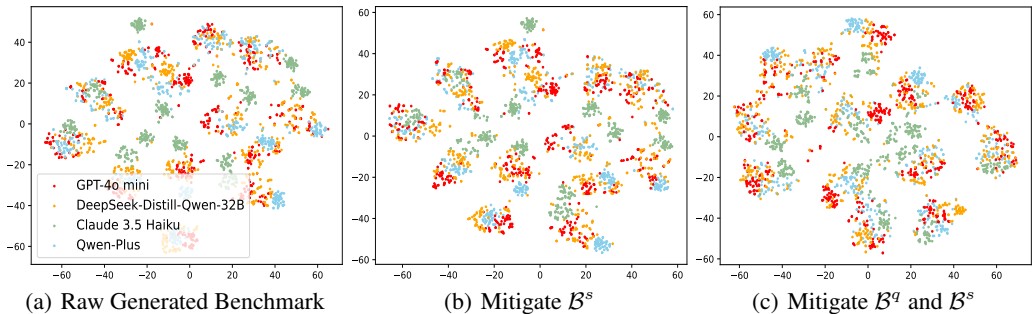

(a) Raw Generated Benchmark     (b) Mitigate $\mathcal{B}^s$     (c) Mitigate $\mathcal{B}^q$ and $\mathcal{B}^s$

Figure 3: T-SNE visualization results of the raw generated benchmarks (a), those with mitigated language style bias (b), and those with both language style and question domain biases suppressed (c) on the Language Understanding task. The presence of 13 clusters is probably due to the task comprising 13 sub-tasks. We use text-embedding-ada-002 (OpenAI, 2024c) as the embedding model.

**Benchmark Diversity.** By expanding the question domain and improving the monotonicity of language style, we observe a clear improvement in the diversity of benchmarks. As shown in the Table 3, after applying SILENCER, the average word frequency entropy (Montahaei et al., 2019) increases from 9.749 to 10.078, and the 2-gram count (Li et al., 2016) rise from 33,560 to 36,932.

**Label Accuracy.** Besides suppressing $\mathcal{B}^l$, we further examine the impact of the Label Calibration strategy on label accuracy. We use OpenAI o3-mini as the judge model to assess label correctness. Experimental results in Table 3 show that Label Calibration can also improve the label accuracy from 0.877 to 0.892, possibly because the model corrects some of its erroneous labels based on right predictions from other generators. This effect further promotes improved evaluation effectiveness.

**Ensemble Weight.** Finally, we calculate the average Pearson correlation between the reciprocal of the ensemble weights estimated by SILENCER and the self-biases across all settings. The results show a high correlation of 0.8252, validating that SILENCER, as expected, assigns greater weights to benchmarks with less self-bias, thereby leading to a higher-quality ensembled benchmark.

Table 3: The impact of the three proposed sample-level strategies for mitigating sub-biases on benchmark diversity (entropy and 2-gram) and label accuracy.

| GENERATORS | ENTROPY↑ | | 2-GRAM↑ | | LABEL ACCURACY↑ | |
|---|---|---|---|---|---|---|
| | RAW | w. SILENCER | RAW | w. SILENCER | RAW | w. SILENCER |
| DEEPSEEK-DISTILL-QWEN-32B | 9.691 | 9.828 | 25493 | 24747 | 0.855 | 0.862 |
| QWEN-PLUS | 9.307 | 10.149 | 25488 | 36707 | 0.819 | 0.846 |
| GPT-4O MINI | 9.610 | 9.931 | 23231 | 25742 | 0.912 | 0.927 |
| CLAUDE 3.5 HAIKU | 9.274 | 9.989 | 14172 | 25286 | 0.833 | 0.884 |
| GPT-4O | 9.942 | 10.023 | 55440 | 56010 | 0.893 | 0.899 |
| GEMINI 2.0 FLASH | 10.300 | 10.363 | 56985 | 53378 | 0.900 | 0.891 |
| QWQ-32B | 10.120 | 10.269 | 34111 | 36657 | 0.927 | 0.936 |
| **AVERAGE** | 9.749 | **10.078** | 33560 | **36932** | 0.877 | **0.892** |

Table 4: The impact of generator number $T$.

| METRICS | $T=3$ | $T=4$ | $T=5$ | $T=6$ | $T=7$ |
|---|---|---|---|---|---|
| NAIVE ENSEMBLE: SELF BIAS $\mathcal{B}$ | 0.031 | 0.020 | 0.015 | 0.011 | 0.008 |
| NAIVE ENSEMBLE: EVALUATION EFFECTIVENESS $r_p$ | 0.909 | 0.920 | 0.925 | 0.932 | 0.937 |
| REWEIGHTED ENSEMBLE: SELF BIAS $\mathcal{B}$ | 0.030 | 0.016 | 0.010 | 0.008 | 0.005 |
| REWEIGHTED ENSEMBLE: EVALUATION EFFECTIVENESS $r_p$ | 0.912 | 0.936 | 0.949 | 0.951 | 0.955 |
| REWEIGHTED ENSEMBLE: WEIGHT ESTIMATION ACCURACY $r_p$ | 0.453 | 0.734 | 0.884 | 0.922 | 0.948 |

## 5.4 Impact of Setting Variants

**Generator Number $T$.** SILENCER primarily suppresses self-bias by leveraging mutual bias neutralization among different generators. Therefore, we examine the effect of varying the generator number $T$. As shown in the Table 4, with a fixed ensembled benchmark size $N$, both Naive Ensemble and Reweighted Ensemble strategies lead to a reduction in $\mathcal{B}$ as $T$ increases. This suggests that using more generators can introduce greater bias diversity, thereby better diluting self-bias and improving Evaluation Effectiveness $r_p$. We also observe that a larger $T$ results in a higher consistency between the reciprocal of the estimated ensemble weight and self-bias (the last row), indicating that the proposed Reweighted Ensemble benefits more from scaling $T$ than Naive Ensemble.

**Benchmark Size $N$.** We also investigate the effect of scaling the benchmark size $N$. As $N$ increases ($50 \rightarrow 100 \rightarrow 200$), the self-bias $\mathcal{B}$ under the Naive Ensemble strategy remains nearly unchanged (0.021), while under the Reweighted Ensemble strategy, $\mathcal{B}$ shows a consistent decline ($0.014 \rightarrow 0.012 \rightarrow 0.011$). We hypothesize that the Reweighted Ensemble benefits from a larger $N$ because more samples enable better estimation of model performance, thereby improving the precision of ensemble weight estimation as we have observed: $0.788 \rightarrow 0.820 \rightarrow 0.839$.

## Conclusions

In this work, we systematically analyze and mitigate the Self-Bias of LLM-as-Benchmark-Generator. Specifically, we validate the existence of Self-Bias and attribute it to three sub-biases. Based on this, we propose SILENCER, a framework that helps generate high-quality, self-bias-silenced benchmarks from multiple levels. We validate the effectiveness and generalizability of SILENCER through experiments across various settings. We further explore the finer-grained impacts of SILENCER on the generated benchmarks and provide insights into the setting variants selection.

**Limitations.** To mitigate self-bias and attain higher-quality benchmarks, our SILENCER framework incurs approximately 30% additional token costs. However, we believe this overhead is worthwhile because (1) users generally prioritize the accuracy of evaluation results over cost, and (2) benchmark sizes used in practice are typically not very huge. For example, we often use MATH-500 (Lightman et al., 2024) instead of the full MATH benchmark. In such cases, the extra cost is acceptable.

**Broader Impacts.** We believe that the SILENCER framework help mitigate potential biases in synthetic data, thereby reducing the risk of distributional shifts associated with internet-based corpora.

**Acknowledgments**

This work is supported by Beijing Natural Science Foundation (No.4222037, L181010).

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

# A  LLM-as-Benchmark-Generator Baselines

We introduce the LLM-as-Benchmark-Generator baselines we used in the experiments as follows:

## A.1  AttrPrompt

AttePrompt (Yu et al., 2023) is a widely used data synthesis method. Its core idea is to first prompt a generator model to produce a set of task-related attributes, each associated with several values; then, based on this, a random set of (attribute, value) pairs is selected each time as constraints to guide the model in generating samples that satisfy both task and attribute constraints.

## A.2  BenchMaker

The BenchMaker (Yuan et al., 2025) method aims to generate high-quality benchmarks by optimizing multiple attributes such as diversity, accuracy, and difficulty. For a given task description, sample diversity is enhanced through attribute-driven generation and an in-batch diversity boosting strategy; benchmark difficulty and controllability are improved via a difficulty diffusion mechanism and strategy guidance; sample accuracy is elevated through stepwise self-correction and conflict-guided contrastive discrimination. Experiments demonstrate that this method produces high-quality benchmarks that are well-aligned with human annotations. In our experiments, we adopted the task descriptions following Yuan et al. (2025). Due to the use of a large number of generators, we used a benchmark size of 50 per sub-task instead of the 500 used in BenchMaker.

# B  Data Contamination Issue Identified in Our Preliminary Experiments

In our preliminary experiments, we observed that some models exhibited relatively better performance on the human-annotated benchmark compared to most (or even all) model-generated benchmarks, resulting in a general negative evaluation bias, as shown in Figure 3. We hypothesize that this may be due to data leakage specific to the human-annotated benchmark. To verify this, we employed the Benchmark-Specific Contamination detection method proposed by ConStat (Dekoninck et al., 2024), which infers the expected performance on the human-annotated benchmark by fitting the performance distributions of various models on synthetic benchmarks. The deviation between the expected and actual performance is used to estimate the degree of contamination. On the Math Reasoning task, ConStat indicated statistically significant contamination (p-value < 0.05) for DeepSeek-Distill-Qwen-32B ($M_4$) and QwQ-32B ($M_6$). On the Language Understanding task, significant contamination was detected for DeepSeek-Distill-Qwen-32B ($M_4$), GPT-4o ($M_5$), QwQ-32B ($M_6$), and Gemini 2.0 Flash ($M_7$). These results align with the models exhibiting consistent negative evaluation bias in Figure 3, supporting our hypothesis.

# C  Proof Section

## C.1  Self-Labeling Inflates Expected Accuracy

**Setup.** Suppose we have $T$ models, denoted by $\{M_1, M_2, \ldots, M_T\}$, each mapping an input $x$ to a probability distribution over possible outputs in a label space $\mathcal{Y}$. For each model $M_i$, let

$$p_i(y \mid x) \;=\; \Pr(\text{model } M_i \text{ outputs } y \mid x), \quad y \in \mathcal{Y}.$$

We examine two expected accuracy measures:

(1) *Self-labeling expectation $E_1$*: the case where each model both generates the label and predicts it:

$$E_1 \;=\; \frac{1}{T} \sum_{i=1}^{T} \sum_{y \in \mathcal{Y}} \underbrace{p_i(y \mid x)}_{\boldsymbol{label}} \underbrace{p_i(y \mid x)}_{\boldsymbol{predict}} \;=\; \frac{1}{T} \sum_{i=1}^{T} \sum_{y \in \mathcal{Y}} \bigl[p_i(y \mid x)\bigr]^2.$$

(2) *Cross-labeling expectation $E_2$*: the case where a model $M_j$ generates the label and another model $M_i$ predicts it:

$$E_2 \;=\; \frac{1}{T^2} \sum_{i=1}^{T} \sum_{j=1}^{T} \sum_{y \in \mathcal{Y}} \underbrace{p_j(y \mid x)}_{label} \, \underbrace{p_i(y \mid x)}_{predict}.$$

**Claim.** We aim to prove that $E_1 \geq E_2$.

**Proof.** Subtract $E_2$ from $E_1$:

$$E_1 - E_2 \;=\; \frac{1}{T} \sum_{i=1}^{T} \sum_{y \in \mathcal{Y}} \big[ p_i(y \mid x) \big]^2 \;-\; \frac{1}{T^2} \sum_{i=1}^{T} \sum_{j=1}^{T} \sum_{y \in \mathcal{Y}} p_j(y \mid x) \, p_i(y \mid x).$$

Rewriting this via a common factorization yields

$$E_1 - E_2 \;=\; \frac{1}{2 \, T^2} \sum_{i=1}^{T} \sum_{j=1}^{T} \sum_{y \in \mathcal{Y}} \Big[ \big( p_i(y \mid x) \big)^2 + \big( p_j(y \mid x) \big)^2 - 2 \, p_i(y \mid x) \, p_j(y \mid x) \Big]$$

$$=\; \frac{1}{2 \, T^2} \sum_{i=1}^{T} \sum_{j=1}^{T} \sum_{y \in \mathcal{Y}} \Big[ p_i(y \mid x) \;-\; p_j(y \mid x) \Big]^2 \;\geq\; 0.$$

Hence

$$E_1 \;-\; E_2 \;\geq\; 0 \quad \Longrightarrow \quad E_1 \;\geq\; E_2.$$

This proves that using each model as its own label generator cannot be worse (in terms of expected accuracy) than having labels generated by a different model.

### C.2 Convergence of the Proposed Bias-Neutralizing Ensemble Algorithm

**Setup.** Let $X \in [0, 1]^{N \times N}$ be a non-negative matrix (the performance matrix in our scenario) whose columns are $x_1, x_2, \ldots, x_N \in \mathbb{R}^N$. We use $N$ to denote the number of generators to avoid confusion with the transpose symbol. Consider an ensemble-weight vector $\alpha = (\alpha_1, \ldots, \alpha_N)^\top \in \Delta^{N-1}$, where

$$\Delta^{N-1} := \big\{ z \in \mathbb{R}^N : z_i \geq 0 \; \forall i, \; \sum_{i=1}^{N} z_i = 1 \big\}.$$

For $u, v \in \mathbb{R}^N$, let

$$\mathrm{corr}(u, v) = \frac{\sum_{k=1}^{N} (u_k - \bar{u})(v_k - \bar{v})}{\sqrt{\sum_{k=1}^{N} (u_k - \bar{u})^2} \sqrt{\sum_{k=1}^{N} (v_k - \bar{v})^2}}, \quad \bar{u} = \frac{1}{N} \sum_{k=1}^{N} u_k, \; \bar{v} = \frac{1}{N} \sum_{k=1}^{N} v_k.$$

**Iterative process.** Fix $\delta > 0$. Define $F^* : \Delta^{N-1} \to \Delta^{N-1}$ by

$$\bar{x} = X\alpha,$$
$$\phi_i(\alpha) = \mathrm{ReLU}\big( \mathrm{corr}(\bar{x}, x_i) \big) + \delta, \qquad i = 1, \ldots, N,$$
$$\alpha_i^{\mathrm{new}} = \frac{\phi_i(\alpha)}{\sum_{j=1}^{N} \phi_j(\alpha)}, \qquad i = 1, \ldots, N,$$

(14)

where $\mathrm{ReLU}(t) = \max\{0, t\}$.

### 1. Existence of a fixed point

**Lemma 1** (Existence). *There exists at least one $\alpha^* \in \Delta^{N-1}$ with $F^*(\alpha^*) = \alpha^*$.*

*Proof.* The simplex $\Delta^{N-1}$ is compact and convex. The map $F^*$ is continuous because (i) $\bar{x} = X\alpha$ depends linearly on $\alpha$, (ii) $\mathrm{corr}(\bar{x}, x_i)$ is continuous in $\bar{x}$ (outside degenerate constant vectors), and (iii) ReLU is continuous and 1-Lipschitz on $\mathbb{R}$. Hence $F^*$ is a continuous self-map on a compact convex set, and Brouwer's fixed-point theorem guarantees at least one fixed point. $\square$

## 2. Uniqueness and contraction

**Theorem 1** (Uniqueness & Contraction). *With $\delta > 0$, the map $F^*$ in (14) is a strict contraction on $\Delta^{N-1}$ under the $\ell^1$ norm. Consequently, the fixed point in Lemma 1 is unique.*

*Sketch.* Take $\alpha, z \in \Delta^{N-1}$ and set $c_i(\alpha) = \mathrm{corr}(X\alpha, x_i),\; c_i(z) = \mathrm{corr}(Xz, x_i)$. Because ReLU is 1-Lipschitz,

$$|\phi_i(\alpha) - \phi_i(z)| = \big|\mathrm{ReLU}(c_i(\alpha)) - \mathrm{ReLU}(c_i(z))\big| \leq |c_i(\alpha) - c_i(z)|.$$

The mapping $\alpha \mapsto X\alpha$ is linear, while the correlation is Lipschitz on the bounded set $[0,1]^N$. Thus there is $L < \infty$ with $\|\phi(\alpha) - \phi(z)\|_1 \leq L\|\alpha - z\|_1$.

Now decompose the update as

$$F^*(\alpha) = (1 - \lambda_\alpha)\frac{\mathbf{1}}{N} + \lambda_\alpha \frac{\phi(\alpha) - \delta\mathbf{1}}{\|\phi(\alpha) - \delta\mathbf{1}\|_1}, \quad \lambda_\alpha = \frac{\|\phi(\alpha) - \delta\mathbf{1}\|_1}{\|\phi(\alpha) - \delta\mathbf{1}\|_1 + N\delta} \in [0,1).$$

Because $\delta > 0$, we have $\sup_\alpha \lambda_\alpha =: q < 1$. A direct calculation using the above representation yields $\|F^*(\alpha) - F^*(z)\|_1 \leq q\|\alpha - z\|_1$. Hence $F^*$ is a strict contraction, so it admits exactly one fixed point. $\square$

## 3. Global convergence to the unique fixed point

**Theorem 2** (Global convergence). *Let $\alpha^{(0)} \in \Delta^{N-1}$ and define $\alpha^{(k+1)} = F^*(\alpha^{(k)})$. There exists a unique $\alpha^* \in \Delta^{N-1}$ with $F^*(\alpha^*) = \alpha^*$, and*

$$\|\alpha^{(k)} - \alpha^*\|_1 \leq q^k \|\alpha^{(0)} - \alpha^*\|_1, \qquad 0 < q < 1,$$

*so the convergence is geometric.*

*Proof.* By Theorem 1, $F^*$ is a contraction on a complete metric space, hence Banach's fixed-point theorem applies directly. $\square$

**Remark.** If for some iterate all correlations are non-positive, ReLU sets them to 0, so $\phi(\alpha) \equiv \delta\mathbf{1}$ and $F^*$ maps *every* point to the uniform distribution. In that (rare) event the process converges in a single step, which is even faster than the geometric rate guaranteed above.

# D   Reference Model List

Apart from the seven generator LLMs, we also include the following LLMs in our reference model set:

1. **doubao-1-5-pro-32k-250115.** https://www.volcengine.com/product/doubao

2. **DeepSeek-V3-0324.** https://huggingface.co/deepseek-ai/DeepSeek-V3-0324

3. **doubao-1-5-lite-32k-250115.** https://www.volcengine.com/product/doubao

4. **DeepSeek-R1-Distill-Llama-70B.** https://huggingface.co/deepseek-ai/DeepSeek-R1-Distill-Llama-70B

5. **Qwen2.5-72B-Instruct.** https://huggingface.co/Qwen/Qwen2.5-72B-Instruct

6. **Llama-3.1-70B-Instruct.** https://huggingface.co/meta-llama/Llama-3.1-70B-Instruct

7. **DeepSeek-R1-Distill-Llama-8B.** https://huggingface.co/deepseek-ai/DeepSeek-R1-Distill-Llama-8B

8. **Athene-70B.** https://huggingface.co/Nexusflow/Athene-70B

9. **gemma-2-27b-it.** https://huggingface.co/google/gemma-2-27b-it

# E  Prompt List

**Prompt for Attribute Integration**

```
Your task is to help create a benchmark with multiple-choice questions
    about "{{ability}}". Specifically:

#### ***Task*** {{task content}}
#### ***Question Content*** {{task content analysis}}
#### ***Options*** {{option analysis}}

Below is a set of attributes and their corresponding values.

{{attributes}}

step 1. Please evaluate them sequentially on two dimensions:

Dimension 1: Assess whether the attribute is closely related to the
    given task and whether it serves as an appropriate attribute for
    controlling the diversity of generated sample.
Dimension 2: Evaluate whether the values corresponding to the
    attribute are comprehensive. If not, suggest any additional
    revisions that should be included.

step 2. Based on this evaluation, please consolidate the high-quality
    attributes and their values. The consolidation process should
    include the following steps:
    1. For attributes with high redundancy, merge them (particularly
       focusing on merging their values).
    2. Provide the optimized attributes and their corresponding values
       .

step 3. Finally, please output all the attributes that you consider
    relevant and meaningful for evaluating the given task along with
    their values. The output should include no fewer than 8 attributes
    , and the more, the better.

Please present the final output of step 3 in the following format (Use
    '###The Final Attributes###' to separate this part from the
    previous analyses and end with ###End of Final Attributes###):

###The Final Attributes###
{Name of Attribute 1}: {Value 1}##{Value 2}##{Value 3}...
(e.g., Reasoning Steps: 1-2##3-4##5-6##>6)
{Name of Attribute 2}: {Value 1}##{Value 2}##{Value 3}...
... ...
{Name of Attribute N}: {Value 1}##{Value 2}##{Value 3}...
###End of Final Attributes###
```

**Prompt for Cross Paraphrase**

```
You are an assistant skilled in rewriting samples. Given a sample:
### Sample Start ###
Question: {input question content}
Options: {input options content}
Label: {input correct option}
### Sample End ###

Your task is to execute the following process for the given sample:
Step 1: Analyze the question and understand the corresponding correct
    answer.
Step 2: Think about how to rewrite the question while keeping the
    original meaning of the question, options, and label unchanged.
    Focus on language style transformation or rephrasing (e.g.,
```

```
    synonym substitution), but ensure the correctness of the rewritten
    sample remains intact.
Step 3: Output the rewritten question in the following format (you can
    also refer to the template below):

Template:
### Sample Start ###
Question:
{question content}
Options:
{options content}
Label:
{correct option}
### Sample End ###
```

**Prompt for Label Calibration**

```
Question:
{{question}}

Candidate Responses:
{{candidates}}

Your task is conduct the steps below strictly:
1. Read the given Question
2. Analyze all the given Candidate Responses One by One
3. Based on the given Candidate Responses, determine the correct
    answer through comprehensive analysis
4. Output your final answer in the format: **My answer is ###{{option
    }}###**
```

