# OpenReview forum: "Silencer: From Discovery to Mitigation of Self-Bias in LLM-as-Benchmark-Generator"
_NeurIPS.cc/2025/Conference — NeurIPS 2025 poster_

### Official Review · Reviewer_wEJW · 2025-07-01

**Clarity:** 3
**Significance:** 3
**Originality:** 3
**Rating:** 4
**Confidence:** 4

**Summary:**

This paper focuses on investigating a timely problem with LLM-generated benchmarks, specifically the self-bias of LLM-as-Benchmark-Generator. In this phenomenon, LLMs tend to achieve overestimated performance on benchmarks they generate themselves. To tackle this problem, the paper first formally defines and quantifies self-bias and empirically demonstrates its prevalence across multiple LLMs and evaluation tasks. Sequentially, the paper heuristically decouples self-bias into three contributors with different relative contributions: question domain bias, language style bias, and wrong label bias. Finally, the paper proposes a new framework, SILENCER, for benchmark generation to address these issues, which mitigates the self-bias at both the sample and benchmark levels. Experimental results on three tasks have validated the effectiveness and generalizability of SILENCER.

**Questions:**

1. Missing reference:
   - AutoBencher: Towards Declarative Benchmark Construction. ICLR 2025.

**Ethical Concerns:**

["NO or VERY MINOR ethics concerns only"]

**Final Justification:**

Thanks for the author's response. I think the additional discussion improves the clarity of Silencer. I decide to maintain my score and still be positive about this paper.

**Limitations:**

See above.

**Quality:**

3

**Strengths And Weaknesses:**

**Pros:**
1. The problem addresses a highly practical and timely issue of NLG evaluation: despite the recent emergence of using LLMs as benchmark generators, potential biases in this paradigm remain underexplored. This paper provides valuable insights into possible biases of LLM-as-Benchmark-Generator and corresponding mitigation strategies.
2. Overall, the paper is well-written, with clear and logical progression from problem identification and analysis to solution development. Each stage is supported by relevant experiments, effectively substantiating its claims.
3. The paper conducts extensive experiments, including different models, tasks, and ablation studies. The experimental results sufficiently demonstrate the effectiveness of the proposed SILENCER framework.

**Cons:**
1. As shown in Figure 1, the self-bias of LLM-as-Benchmark-Generator varies substantially across different base models and tasks. A detailed analysis of these differences would greatly enhance the paper. For instance, the two reasoning models, DeepSeek-Distill-Qwen-32B and QwQ-32B, appear to consistently exhibit the lowest levels of self-bias.  In addition to self-bias, there appears to be evaluation bias between different models as well  (for example, between M1 and M7 in the Math Reasoning task). It would be valuable for the authors to further investigate whether the SILENCER framework is effective in mitigating cross-model evaluation bias.
2. The decomposition of self-bias in Section 3.2 seems a little ad hoc. The authors could further clarify the completeness and rationality of the sub-bias decomposition empirically or theoretically.
3. For Section 4.3, considering that the Bias-Neutralizing Ensemble Algorithm is an iterative algorithm, it would be beneficial for the authors to analyze the number of convergence steps and the evolution of self-bias during the iterative process in their experiments.

---

> ### Author Rebuttal · Authors · 2025-07-30
>
> **Dear Reviewer wEJW, thank you for recognizing the significance of our work, the clarity of our writing, and the effectiveness of our method. Also, we sincerely appreciate you taking the time to read our manuscript and provide valuable feedback. Below are our responses to the concerns you raised, which will be incorporated into the updated version to further enhance the quality of our submission.**
>
> ---
> > **`As shown in Figure 1, the self-bias of LLM-as-Benchmark-Generator varies substantially across different base models and tasks. A detailed analysis of these differences would greatly enhance the paper. For instance, the two reasoning models, DeepSeek-Distill-Qwen-32B and QwQ-32B, appear to consistently exhibit the lowest levels of self-bias. In addition to self-bias, there appears to be evaluation bias between different models as well (for example, between M1 and M7 in the Math Reasoning task). It would be valuable for the authors to further investigate whether the SILENCER framework is effective in mitigating cross-model evaluation bias.`**
>
> We appreciate your insightful suggestions, and indeed, the two phenomena you mentioned also puzzled us during our experiments: (1) $\mathcal{M}\_4$ and $\mathcal{M}\_6$ consistently exhibited the lowest levels of self-bias across multiple tasks; (2) in the Math Reasoning task, $\mathcal{M}\_1$ and $\mathcal{M}\_7$ consistently showed higher evaluation bias.
>
> Through our subsequent systematic analysis, we identified the underlying cause of these phenomena as the **data contamination issue , as described in lines 97–100 of our paper**. Using the ConStat method proposed in [1], we verified in Appendix B (lines 747–752) that, on the MATH dataset, both DeepSeek-Distill-Qwen-32B ($\mathcal{M}\_4$) and QwQ-32B ($\mathcal{M}\_6$) face statistically significant risks of data contamination (p-value < 0.05). This indicates that the performance and relative performance of $\mathcal{M}\_4$ and $\mathcal{M}\_6$ measured on the human-annotated benchmark are overestimated (e.g., $R^{\text{observed}}(\mathcal{M}\_4|\mathcal{M}^{\text{human}},\mathcal{M}^{\text{human}},\mathcal{M}^{\text{ref}}\_{1:K})$). According to the formula for relative performance (Equation 1):
>
> $$
> R(\mathcal{M}|\mathcal{M}^{\text{que}},\mathcal{M}^{\text{lab}},\mathcal{M}^{\text{ref}}\_{1:K})=\frac{K \cdot R(\mathcal{M}|\mathcal{M}^{\text{que}},\mathcal{M}^{\text{lab}})}{\sum\_{k=1}^K R(\mathcal{M}\_k^{\text{ref}}|\mathcal{M}^{\text{que}},\mathcal{M}^{\text{lab}})}
> $$
>
> we can see that these inflated performances indirectly lead to lower relative performance for other models (such as the $\mathcal{M}\_1$ and $\mathcal{M}\_7$ you mentioned) on the human-annotated benchmark (e.g., $R^{\text{observed}}(\mathcal{M}\_1|\mathcal{M}^{\text{human}},\mathcal{M}^{\text{human}},\mathcal{M}^{\text{ref}}\_{1:K})$). Furthermore, according to Equation 2:
>
> $$
> \mathcal{B}(\mathcal{M}|\mathcal{M}^{\text{gen}}) = R(\mathcal{M}|\mathcal{M}^{\text{gen}}, \mathcal{M}^{\text{gen}},\mathcal{M}^{\text{ref}}\_{1:K}) - R(\mathcal{M}|\mathcal{M}^{\text{human}}, \mathcal{M}^{\text{human}},\mathcal{M}^{\text{ref}}\_{1:K})
> $$
>
> we can see that $\mathcal{B}(\mathcal{M}\_4|\mathcal{M}^{\text{gen}})$ will be underestimated since $R^{\text{observed}}(\mathcal{M}\_4|\mathcal{M}^{\text{human}},\mathcal{M}^{\text{human}},\mathcal{M}^{\text{ref}}\_{1:K})$ is overestimated, while $\mathcal{B}(\mathcal{M}\_1|\mathcal{M}^{\text{gen}})$ will be overestimated since $R^{\text{observed}}(\mathcal{M}\_1|\mathcal{M}^{\text{human}},\mathcal{M}^{\text{human}},\mathcal{M}^{\text{ref}}\_{1:K})$ is underestimated, which is consistent with the phenomena observed by both you and us in Figure 1. The same explanation applies to the observations in the Language Understanding task, as detailed in Appendix B.
>
> Thank you again for your careful reading. We will highlight and clarify this part in the revised version of the paper.
>
> ---
> > **`The decomposition of self-bias in Section 3.2 seems a little ad hoc. The authors could further clarify the completeness and rationality of the sub-bias decomposition empirically or theoretically.`**
>
> Thank you very much for your suggestion. We will clarify the rationale and completeness of decomposing the self-bias of LLM-as-Benchmark-Generator into question domain sub-bias, wrong label sub-bias, and language style sub-bias from two perspectives.
>
> **First, regarding the rationale**, two empirical studies in the education field investigating students making their own test papers [2,3] provide supporting evidence:
>
> - Students tend to focus on “topics they consider important or are good at”; open-ended responses in [2] frequently mention “being able to focus on what I think are the most important points,” which leads to higher grades. At the same time, [3] also notes that student-generated questions may fail to cover all knowledge points, confirming that students making their own questions introduces a question domain bias. **These findings support the rationality of our defined question domain sub-bias**.
> - Students prefer to organize the questions and answers using their own language style, believing that it is “fairer and better showcases learning,” and helps to “relieve anxiety and unease,” allowing them to perform at a higher level [2,3]. **This evidence supports the rationality of our defined language style sub-bias**.
> - [2] points out that in human scenarios, it is necessary for teachers to grade accuracy to mitigate the bias from “self-determined answers,” but there can still be ceiling effects if grading is too lenient. They specifically use “accuracy” and “breadth” metrics to counter the risks of “only setting easy questions or giving favorable answers.” [3] also notes that teachers must specifically revise the answers to questions written by students to ensure fairness in grading. **This is fully consistent with our extrapolation of the wrong label sub-bias**: without dedicated mitigation, correlated errors from the same source will translate into higher apparent scores.
>
> **Regarding completeness**, we note that the question set determines what content is assessed, the language style determines the presentation, and together these determine what the tester actually sees. The label set determines the tester’s final score. Therefore, the question set, language style, and label set together constitute a complete benchmark. Based on this, we consider potential sub-biases from these three aspects, making our decomposition of sub-biases comprehensive and complete.
>
> ---
> > **`For Section 4.3, considering that the Bias-Neutralizing Ensemble Algorithm is an iterative algorithm, it would be beneficial for the authors to analyze the number of convergence steps and the evolution of self-bias during the iterative process in their experiments.`**
>
>
> Thank you for your comprehensive suggestion. In response, for the Language Understanding task, we have supplemented our analysis by tracking four metrics throughout the iterative process: self-bias $\mathcal{B}$, the evaluation effectiveness $r\_p$ of the integrated benchmark, the convergence criterion $\sum\_{i=1}^T |\alpha\_i^{\text{new}}-\alpha\_i|$, and the proportion of cases that achieved convergence. The changes in these metrics across iteration steps are shown in the table below:
>
> | Iteration Step | $\mathcal{B}$ |Pearson Correlation $r\_p$| $\sum\_{i=1}^T \|\alpha\_i^{\text{new}}-\alpha\_i\|$|Convergence Rate %|
> |-|:-:|:-:|:-:|:-:|
> |0|0.0076|0.7953|3.36|0|
> |1|0.0042|0.8237|2.43|0|
> |2|0.0029|0.8285|0.22|0|
> |3|0.0021|0.8320|0.11|0|
> |4|0.0017|0.8320|0.07|0|
> |5|0.0015|0.8335|0.05|0.3|
> |18.57|0.0011|0.8340|<1e-6|100|
>
> Here, step=0 corresponds to the scenario where benchmarks generated by all generators are integrated with equal weights. As shown, in the early stages of iteration (the first five steps), both the self-bias $\mathcal{B}$ and the evaluation effectiveness $r_p$ are significantly optimized as the convergence criterion $\sum\_{i=1}^T |\alpha_i^{\text{new}}-\alpha\_i|$ rapidly decreases. Meanwhile, since our convergence threshold is set at $\sum\_{i=1}^T |\alpha_i^{\text{new}}-\alpha\_i|<1\text{e}{-6}$, only 0.3% of cases had converged by the end of the fifth step. On average, it takes 18.57 iterations to reach convergence. Although the marginal benefit from further iterations gradually diminishes, given that the time consumed by each iteration is almost negligible, we adopted the current settings.
>
> ---
> > **`Missing reference: AutoBencher: Towards Declarative Benchmark Construction. ICLR 2025.`**
>
> Thank you for pointing out the additional related work. AutoBencher [4] is also an LLM-as-Benchmark-Generator approach (powered by GPT-4), and we will include this method in the Related Work section of the revised version.
>
>
> **References**
>
> [1] Constat: Performance-based323contamination detection in large language models
>
> [2] “Exams by You”: Having Students Write and Complete Their Own Exams During the COVID-19 Pandemic
>
> [3] Asking the right questions: Using student-written exams as an innovative approach to learning and evaluation
>
> [4] AutoBencher: Towards Declarative Benchmark Construction

---

> > ### Comment · Reviewer_wEJW · 2025-08-04
> >
> > Thanks for your response. I think the additional discussion improves the clarity of Silencer. I decide to maintain my score and still be positive about this paper.

---

### Official Review · Reviewer_iMmg · 2025-07-03

**Clarity:** 4
**Significance:** 4
**Originality:** 4
**Rating:** 5
**Confidence:** 4

**Summary:**

The paper discusses the phenomenon of self-bias in large language models (LLMs) used as benchmark generators, proposing a framework called SILENCER to mitigate this bias and improve evaluation accuracy. The framework includes strategies to mitigate sub-biases at both the sample and benchmark levels. ​ SILENCER employs three main strategies at the sample level:

- Attribute Integration to create domain-unbiased question sets. ​
- Cross Paraphrase to enhance linguistic diversity and reduce language style bias. ​
- Label Calibration to improve label accuracy and reduce wrong label bias. ​

At the benchmark level, a Bias-Neutralizing Ensemble algorithm is used to weigh benchmarks based on their self-bias, improving overall evaluation effectiveness.

SILENCER reduces self-bias to near zero, achieving an average improvement in Pearson correlation from 0.655 to 0.833 with high-quality human-annotated benchmarks. ​The framework is tested across three tasks and seven LLM generators, confirming its generalizability and  effectiveness. Detailed ablation studies validate the contribution of each component in SILENCER, providing insights into the impact of factors like the number of generators and benchmark size on evaluation effectiveness.

**Questions:**

Is there any evaluation on the paraphrases generated wrt quality and correctness? Overall, qualitative assesments are lacking in the paper and could be interesting to look at.

**Ethical Concerns:**

["NO or VERY MINOR ethics concerns only"]

**Final Justification:**

No changes based on author discussion. All limitations I highlighted were addressed.

**Limitations:**

yes

**Paper Formatting Concerns:**

There is a NeurIPS checklist at the end.

**Quality:**

4

**Strengths And Weaknesses:**

- The paper introduces a novel framework, SILENCER, to mitigate self-bias in LLM-as-Benchmark-Generator methods, addressing an under-explored issue in the field. ​
- The paper provides detailed explanations of the methodology, including formal definitions, experimental settings, and theoretical proofs, ensuring transparency and understanding. ​
- The work has broad implications for improving the reliability of LLM-generated benchmarks, which are increasingly used in AI research and applications.

---

> ### Author Rebuttal · Authors · 2025-07-30
>
> **Dear Reviewer iMmg, thank you for your comprehensive recognition of the motivation, novelty, and significance of our work.
> Also, we sincerely appreciate you taking the time to read our manuscript and provide valuable feedback. Below are our response to the concern you raised, which will be incorporated into the updated version to further enhance the quality of our submission.**
>
> ---
> > **`Is there any evaluation on the paraphrases generated wrt quality and correctness?Overall, qualitative assessments are lacking in the paper and could be interesting to look at.`**
>
>
> We sincerely appreciate your suggestion to conduct a qualitative analysis of the paraphrased questions, as this provides us with a more intuitive understanding of how the items change through paraphrasing. To this end, we randomly selected ten samples generated by DeepSeek-Distill-Qwen-32B from the Language Understanding Task and analyzed the characteristics of the paraphrased items produced by different generator models. Below is one example:
>
> **\<DeepSeek-Distill-Qwen-32B generated one\>**
>
> Question:
> How did the introduction of cash crops in 19th-century African, Caribbean, and Southeast Asian colonies primarily affect their economies and societies?
>
> Options:
>
> A. Enhanced social equality through equitable distribution of cash crop profits.
> B. Increased economic diversification and self-sufficiency in local markets.
> C. Decreased labor availability due to migration to urban areas.
> D. Improved infrastructure development to support cash crop exports.
> E. Enhanced cultural preservation through traditional agricultural practices.
> F. Economic dependency on colonial powers due to export-focused agriculture.
> G. Reduced environmental impact due to sustainable farming practices.
> H. Improved local economies through diversified crop production.
> I. Reduced reliance on colonial powers through export profits.
> J. Increased political autonomy through economic independence.
>
>
> **\<paraphrased by QWQ-32B\>**
>
> Question:
> What were the main impacts on the economies and societies of 19th-century African, Caribbean, and Southeast Asian colonies as a result of the introduction of cash crops?
>
> Options:
>
> A. Improved social equality due to fair sharing of cash crop profits.
> B. Boosted economic diversity and self-reliance in local markets.
> C. Reduced workforce availability as people moved to cities.
> D. Enhanced infrastructure to facilitate cash crop exports.
> E. Strengthened preservation of traditional farming methods.
> F. Economic reliance on colonial powers due to export-oriented agriculture.
> G. Lower environmental effects thanks to sustainable farming.
> H. Economic growth through varied crop production.
> I. Less dependence on colonial powers through export earnings.
> J. Greater political independence through economic self-sufficiency.
>
> - **QWQ-32B’s paraphrase adheres to the principles of minimal alteration and maximum fidelity, without changing the answer direction.**
>
> **\<paraphrased by Qwen-Plus\>**
>
> Question:
> How did the economies and societies of African, Caribbean, and Southeast Asian colonies mainly change as a result of the introduction of cash crops during the 19th century?
>
> Options:
>
> A. Promoted equal social status by fairly sharing earnings from cash crops.
> B. Encouraged a wider variety of economic activities and greater independence in regional markets.
> C. Caused a shortage of workers as people moved toward cities.
> D. Strengthened transportation and communication systems to facilitate the export of cash crops.
> E. Maintained cultural traditions by continuing customary farming techniques.
> F. Created economic reliance on colonial rulers due to a focus on export agriculture.
> G. Lowered environmental damage through eco-friendly farming methods.
> H. Boosted domestic economies by producing a range of crops.
> I. Lessened dependence on colonial authorities by generating income from exports.
> J. Expanded political freedom as a result of economic self-reliance.
>
> - **Qwen-Plus’s paraphrase enhances clarity with a more formal and definitive style; its explanations are more explicit but do not go beyond the intended scope, resulting in high quality.**
>
>
> **\<paraphrased by GPT-4o mini\>**
>
> Question:
> What was the primary impact of cash crop cultivation in the 19th century on the economies and societies of African, Caribbean, and Southeast Asian colonies?
>
> Options:
>
> A. Promoted social equity through fair distribution of cash crop revenues.
> B. Led to greater economic diversification and self-reliance in local markets.
> C. Reduced labor availability as people migrated to urban centers.
> D. Fostered infrastructure improvements to facilitate cash crop exports.
> E. Strengthened cultural preservation via traditional farming methods.
> F. Created economic dependency on colonial authorities due to a focus on export-oriented agriculture.
> G. Minimized environmental damage through sustainable farming techniques.
> H. Boosted local economies through a variety of crop production.
> I. Decreased reliance on colonial powers through profits from exports.
> J. Enhanced political independence through economic self-sufficiency.
> K. Improved literacy rates through agricultural education programs.
>
> - **GPT-4o mini’s paraphrase introduces a new option K, which does not align with our prompt instruction that paraphrasing should only modify the linguistic style without altering the substantive content. However, after examining the other nine cases, we found this to be an isolated occurrence.**
>
> **\<paraphrased by Claude3.5 Haiku\>**
>
> Question:
> Examine the primary socioeconomic transformations induced by the introduction of cash crop economies in colonized territories across Africa, the Caribbean, and Southeast Asia during the 19th century.
>
> Options:
>
> A. Stimulated localized agricultural innovation and technological adaptation.
> B. Promoted horizontal economic integration among indigenous communities.
> C. Generated significant population mobility and demographic shifts.
> D. Catalyzed infrastructural development through colonial investment.
> E. Preserved traditional agricultural methodologies and cultural practices.
> F. Established systemic economic subordination to metropolitan colonial powers.
> G. Minimized ecological disruption through sustainable agricultural strategies.
> H. Expanded agricultural diversity and market complexity.
> I. Facilitated economic emancipation through international trade mechanisms.
> J. Enhanced indigenous political agency through economic transformations.
>
> - **Claude 3.5 Haiku’s paraphrase systematically academicizes both the correct and distractor options, increasing the difficulty and confusion while maintaining correctness.**
>
> **\<paraphrased by GPT-4o\>**
>
> Question:
> What was the primary impact of cash crop cultivation introduced in 19th-century colonies across Africa, the Caribbean, and Southeast Asia on their economies and societies?
>
> Options:
>
> A. Promoted social equity by fairly distributing cash crop revenues.
> B. Fostered economic self-reliance and diversification in domestic markets.
> C. Reduced the availability of labor due to urban migration.
> D. Stimulated infrastructure development to facilitate cash crop exports.
> E. Preserved cultural traditions through indigenous farming methods.
> F. Created economic reliance on colonial powers through export-oriented agriculture.
> G. Minimized ecological damage through sustainable agricultural practices.
> H. Strengthened local economies by introducing a variety of crops.
> I. Lessened dependence on colonial authorities by generating export income.
> J. Advanced political independence through economic self-sufficiency.
>
> - **GPT-4o’s paraphrase demonstrates a natural, balanced, and stable style, without expanding the set of options or altering the difficulty structure.**
>
> **\<paraphrased by Gemini 2.0 Flash\>**
>
> Question:
> What was the principal consequence of introducing cash crop cultivation in 19th-century African, Caribbean, and Southeast Asian colonial territories on their economic systems and societal structures?
>
> Options:
>
> A. It led to greater social equity through the fair distribution of agricultural profits.
> B. It fostered increased economic variety and self-sufficiency within local markets.
> C. It resulted in a decline in available labor due to significant urban migration.
> D. It spurred significant infrastructure enhancements primarily to support export activities.
> E. It reinforced cultural preservation by maintaining traditional farming methods.
> F. It established economic reliance on imperial powers due to an agriculture system centered on exports.
> G. It minimized environmental degradation through the adoption of sustainable farming practices.
> H. It strengthened domestic economies via the diversification of agricultural output.
> I. It decreased dependence on colonial authorities through substantial export earnings.
> J. It advanced political self-governance through the achievement of economic independence.
>
> - **Gemini 2.0 Flash’s paraphrase is characterized by more comprehensive and explanatory narratives.**
>
> **In summary**, each paraphrasing model exhibits a distinct linguistic style, which qualitatively reveals the language style sub-bias. At the same time, through detailed case studies, we found that apart from the single instance with GPT-4o mini introducing a new option (which does not affect the correctness of the question), all paraphrased questions (a total of 6 × 10 = 60 cases) maintained alignment with the original items in both content and assessed capability, demonstrating high quality. This indicates that paraphrasing for language style is well within the capability of current LLMs.
>
> **We will include this discussion in our revised version. We believe this addition will further improve the quality of our work.**

---

### Official Review · Reviewer_sV2H · 2025-07-03

**Clarity:** 3
**Significance:** 2
**Originality:** 3
**Rating:** 5
**Confidence:** 3

**Summary:**

The paper studies the self-bias of LLM generated benchmarks with respect to the same or similar LLMs to be evaluated or trained on the generated data. Typically, an LLM will be biased to overperform on a benchmark generated by itself. The paper classifies this self-bias into three categories: question domain, language style and labeling and proposes techniques using reference models and human annotated datasets to measure these self-biases. The authors show experimentally the significant self-bias of each LLM in a collection of seven popular LLMs from the industry, on Math Reasoning and Language Understanding datasets. Based on these measures, the paper proposes attribute integration, cross paraphrase and label calibration methods respectively to alleviate the three kinds of self-bias, and further proposes a weighted ensemble approach to combine the benchmarks generated by different LLMs in the collection so that the resultant dataset has low self bias. Experiments on 7 industry LLMs (with 9 others to complete the reference set) are conducted using ablations over baseline generator methods, which show that the three types of self-bias are improved using the proposed method.

**Questions:**

1. It will be helpful to provide examples of generator outputs: question, answer and label for the benchmarks generated/used in the experiments.
2. Line 241: define $r_p$ formally.
3. In Table 2, what is the difference between “$\mathcal{B}^q, \mathcal{B}^s, \mathcal{B}^l$ w. Ensemble” and SILENCER ?

**Ethical Concerns:**

["NO or VERY MINOR ethics concerns only"]

**Final Justification:**

The main concern of using o3-mini as a proxy for human annotators has been addressed by the authors, through additional experiments showing that the model has low error rate compared to human annotators, and can therefore serve as a proxy for the latter. Apart from this the authors have included detailed experimental results and examples of the generator outputs. My concerns have largely been addressed.

**Limitations:**

Yes

**Quality:**

3

**Strengths And Weaknesses:**

Strengths:
1. The paper is a fairly comprehensive study of self-bias in LLM generated Benchmarks.
2. The notions and metrics of self-bias introduced by the paper are natural, well reasoned, and also shown to be prevalent in popular models.
3. The proposed self-bias mitigation method combines individual approaches targeting each of the categories of self-bias – along with weighted ensembling – and is shown experimentally to significantly decrease the self bias for several LLMs across two tasks.

Weaknesses:
1. On line 115, the authors state that they use o3-mini as a proxy for human annotators. Could this not make similar models (e.g. those trained on similar datasets) erroneously perform better in terms of self-bias in the experiments.
2. The presented experimental results seem to have a few gaps: (i) only aggregated bias scores over all generator models are included in Tables 1 and 2, instead of those for each of the seven models and, (ii) the intermediate ablations are not included for AttrPrompt baseline method in Table 2, effectively, the SILENCER method is fully evaluated only for BenchMaker.

---

> ### Author Rebuttal · Authors · 2025-07-30
>
> **Dear Reviewer sV2H, thank you for acknowledging the motivation of our work, the comprehensiveness of our research, and the effectiveness of our method. Also, we sincerely appreciate you taking the time to read our manuscript and provide valuable feedback. Below are our responses to the concerns you raised, which will be incorporated into the updated version to further enhance the quality of our submission.**
>
> ---
> > **`On line 115, the authors state that they use o3-mini as a proxy for human annotators. Could this not make similar models (e.g. those trained on similar datasets) erroneously perform better in terms of self-bias in the experiments?`**
>
> We fully understand your concerns, and we greatly appreciate that your question is based on a thorough reading of our paper. Our response is as follows:
>
> In Section 3.2, when quantifying the Question Domain Sub-bias, due to the large scale of the test data (over 1,000 samples), we employed o3-mini as a human proxy to provide labels for the generated questions. We recognize your concern that labels provided by o3-mini may still contain errors, and that such errors might be biased, thereby affecting the measured Question Domain Sub-bias.
>
> To address this, for each sub-task, we randomly sampled 20 generated questions and manually verified the labels given by o3-mini. All human annotators held at least a bachelor's degree and were instructed to mark the answer as "uncertain" if they were unsure about its correctness. In the end, all annotators provided confident validation results. The verification showed that the error rates of o3-mini’s labeling were **7.8%** for Math Reasoning and **6.9%** for Language Understanding tasks.
>
> Furthermore, we examined whether such labeling errors introduce additional bias, as measured by the following formula:
>
> $$\mathcal{B}^{l}\_{\mathcal{M}^{\text{o3-mini}}}(\mathcal{M}) = R(\mathcal{M}|\mathcal{M}, \mathcal{M}^{\text{o3-mini}},\mathcal{M}^{\text{ref}}\_{1:K}) - R(\mathcal{M}|\mathcal{M}, \mathcal{M}^{\text{human}},\mathcal{M}^{\text{ref}}\_{1:K})$$
>
> Experimental results show that, on Math Reasoning and Language Understanding tasks, the average values of $ \mathcal{B}^{l}\_{\mathcal{M}^{\text{o3-mini}}}(\mathcal{M}) $ across 7 generators were **0.0014** and **0.0007**, respectively. Compared to the measured values of $ \mathcal{B}^{q}$ (**0.0247** and **0.0077**), the magnitude of this error is negligible and does not affect our experimental conclusions.
>
> **In summary, the results demonstrate that using o3-mini as a human proxy to evaluate sub-biases in LLM-as-benchmark-generator does not impact the conclusions in Section 3**. In fact, this is understandable: models with training trajectories similar to o3-mini may perform better under this setting due to higher homogeneity (indeed, we observed that $\mathcal{B}^{l}\_{\mathcal{M}^{\text{o3-mini}}}(\mathcal{M}^{\text{GPT-4o}})$ was 0.0062 for Math Reasoning), while more distinct models (e.g., $\mathcal{B}^{l}\_{\mathcal{M}^{\text{o3-mini}}}(\mathcal{M}^{\text{Gemini 2.0 Flash}})$ was -0.0051 for Math Reasoning) may perform worse. As a result, these effects nearly offset each other on average, as shown by our experimental results above.
>
> We sincerely thank you for your suggestion. By incorporating this additional analysis, we believe our results are now more comprehensive and convincing.
>
> ---
> > **`The presented experimental results seem to have a few gaps:(i) only aggregated bias scores over all generator models are included in Tables 1 and 2, instead of those for each of the seven models.(ii) the intermediate ablations are not included for AttrPrompt baseline method in Table 2. Effectively, the SILENCER method is fully evaluated only for BenchMaker.`**
>
> Due to space limitations, we did not include these fine-grained experimental results in the main tables, but instead reported the results that support the core conclusions of our study. We fully acknowledge your suggestion; presenting such details would help researchers gain a more comprehensive understanding of the distribution of self-bias and the effectiveness of Silencer from different perspectives. Below are the experimental results you mentioned. We will add these information to the revised version.
>
> **Detailed Results of Table1-Math Reasoning**
>
> | Models | $\mathcal{B}$|$\mathcal{B}^s$|$\mathcal{B}^q$|$\mathcal{B}^l$|
> |-|-|-|-|-|
> |GPT-4o mini|0.0923|-0.0067|0.0064|0.0675|
> |Qwen-Plus|0.0576|0.0162|0.0088|0.0455|
> |Claude 3.5 Haiku|0.3895|0.0357|0.0524|0.2464|
> |DeepSeek-Distill-Qwen-32B|-0.0210|-0.0146|-0.0021|0.0135|
> |GPT-4o|0.0567|0.0028|0.0004|0.0741|
> |QwQ-32B|0.0076|0.0750|0.0731|0.0844|
> |Gemini 2.0 Flash|0.1398|0.0355|0.0341|0.1140|
> |Avg.|0.1032|0.0205|0.0247|0.0921|
>
> **Detailed Results of Table1-Language Understanding**
>
> | Models | $\mathcal{B}$|$\mathcal{B}^s$|$\mathcal{B}^q$|$\mathcal{B}^l$|
> |-|-|-|-|-|
> |GPT-4o mini|0.1230|0.0152|0.0213|0.0765|
> |Qwen-Plus|0.1108|0.0186|0.0127|0.0772|
> |Claude 3.5 Haiku|0.1297|0.0136|0.0167|0.1234|
> |DeepSeek-Distill-Qwen-32B|0.0057|-0.0128|-0.0017|0.0083|
> |GPT-4o|0.0129|0.0092|0.0031|0.0288|
> |QwQ-32B|-0.0060|-0.0090|0.0086|0.0229|
> |Gemini 2.0 Flash|0.0268|0.0082|-0.0066|0.0444|
> |Avg.|0.0575|0.0061|0.0077|0.0545|
>
>
> **Detailed Results of Table2-AttrPrompt Baseline**
>
>  |_|Math|Reasoning|Language|Understanding|Commonsense|Reasoning|
>  |-|:-:|:-:|:-:|:-:|:-:|:-:|
> |Methods| $\mathcal{B}$|$r_p$|$\mathcal{B}$|$r_p$|$\mathcal{B}$|$r_p$|
> |Baseline|0.1427|0.6179|0.1148|0.5843|0.1093|0.6106|
> |w. Attribute Integration:$\mathcal{B}^q$|0.1211|0.6158|0.1058|0.6109|0.1087|0.6149|
> |w. Cross Paraphrase:$\mathcal{B}^s$|0.1169|0.6349|0.1102|0.5953|0.1016|0.6095|
> |w. Label Calibration:$\mathcal{B}^l$|0.1048|0.6508|0.1025|0.6228|0.0988|0.6327|
> |w. $\mathcal{B}^q$ $\mathcal{B}^s$ $\mathcal{B}^l$|0.0957|0.6677|0.1034|0.6341|0.0927|0.6384|
> |w. $\mathcal{B}^q$ $\mathcal{B}^s$ $\mathcal{B}^l$ w. Ensemble|0.0340|0.8255|0.0203|0.7437|0.0257|0.7505|
> |w. Silencer|0.0187|0.8641|0.0125|0.7725|0.0074|0.7762|
>
> The table presents the full evaluation of the SILENCER method based on the AttrPrompt Baseline, providing further validation for the effectiveness of the proposed sub-modules.
>
> It should be noted that for Table 2 of the original manuscript, if we were to present detailed results for each task (3 in total), each baseline method (2 in total), and each experimental setting (7 in total), and report individual bias scores for each model, this would result in a total of 3 × 2 × 7 = 42 tables. Each table would contain 7 rows (for the seven generator models) and 4 columns ($\mathcal{B}$, $\mathcal{B}^q$, $\mathcal{B}^s$, $\mathcal{B}^l$), which would be extremely difficult to present. In practice, we have already visualized these detailed results as figures, which makes them easier to read and analyze. We will include these visualizations in the appendix of the revised version.
>
> ---
> >  **`It will be helpful to provide examples of generator outputs: question, answer and label for the benchmarks generated/used in the experiments.`**
>
> We fully agree with your suggestion. Below are examples of samples generated by the Silencer framework on the two tasks. We will include more cases in the appendix of the revised version.
>
> **Math Reasoning - Algebra**
>
> Question:
>
> An investor divides \\$100 into two parts, $x$ and $y$, invested in two different accounts. The first account offers an annual return modeled by the equation \( R_A = x^2 - 10x + 25 \), and the second account offers a return of \( R_B = y \). The total return after one year is \\$125, and the total investment is \\$100. Determine the amount invested in the first account.
>
> Candidates:
>
> A. \\$5  B. \\$10  C. \\$11  D. \\$15
>
> Label: C
>
> **Commonsense Reasoning**
>
> Question:
>
> In a South Asian city, a lawyer is providing legal advice to a client regarding a delicate family issue that involves a family dispute with relatives. During the consultation, the client becomes clearly distressed, expressing feelings of betrayal and anxiety. Despite the lawyer's attempts to reassure them, the client shows hesitation to proceed with legal action, stating that they now believe the issue can be resolved through family mediation. Considering the lawyer's responsibilities, the cultural value placed on family unity, and the client's emotional state, what is the most likely next step the lawyer will take?
>
> Candidates:
>
> A. Advise the client to seek professional counseling to address their anxiety.
>
> B. Strongly push the client to file a lawsuit immediately, ignoring their wish for mediation.
>
> C. Explain the legal process for family mediation and help the client contact a qualified mediator.
>
> D. Suggest the client abandon legal proceedings entirely to preserve family harmony.
>
> Label:  C
>
> ---
> > **`Line 241: define rp formally.`**
>
> We have formally defined $r_p$ in line 233-235: “In addition to self-bias, we also compute the Pearson correlation $r_p$ of reference models' performance between model-generated and high-quality human-annotated benchmarks to assess the evaluation effectiveness of the generated benchmarks.” It can be calculated as：
>
> $$
> r\_p = \text{PearsonCorrelation}([\text{Performance}(\mathcal{M}^{\text{ref}}\_i\ |\ \text{human-annotated benchmark})]\_{i=1}^K,[\text{Performance}(\mathcal{M}^{\text{ref}}\_i\ |\ \text{model-generated benchmark})]\_{i=1}^K)
> $$
>
> ---
> > **`In Table 2, what is the difference between “$B_q,B_s,B_l$ w. Ensemble” and SILENCER?`**
>
> The distinction is as follows: “$B_q$, $B_s$, $B_l$ w. Ensemble” refers to equally weighted ensembling of benchmarks generated by different generators. In contrast, “Silencer” denotes weighted ensembling of benchmarks generated by different generators, where the weights are set as the inverse of the self-bias magnitude estimated by Algorithm 1. This approach enables further suppression of self-bias at the benchmark level, as reflected in the improvements shown in Table 2 compared to “$B_q$, $B_s$, $B_l$ w. Ensemble”.

---

> > ### Comment · Reviewer_sV2H · 2025-08-05
> >
> > Thank you for your rebuttal, which has largely addressed my concerns. With the additional experiments and explanations the authors have proposed to include, I believe this paper is substantially strengthened.

---

### Official Review · Reviewer_8ACx · 2025-07-21

**Clarity:** 3
**Significance:** 2
**Originality:** 3
**Rating:** 4
**Confidence:** 4

**Summary:**

This paper investigates self-bias in benchmarks generated by large language models (LLMs), where an LLM tends to overestimate its own performance on benchmarks it creates due to alignments in question domains, language styles, and labeling errors. It decouples self-bias into sub-components (question domain bias, wrong label bias, and language style bias) and proposes the SILENCER framework to mitigate it. At the sample level, SILENCER introduces three strategies: Attribute Integration (to diversify question domains), Cross Paragraphs (to vary language styles), and Label Calibration (to correct wrong labels). At the benchmark level, it employs a Bias-Neutralizing Ensemble algorithm that weights benchmarks inversely to their estimated self-bias. The framework is evaluated on three tasks (math reasoning, language understanding, commonsense reasoning), seven LLM generators, and two benchmark generation methods, demonstrating a 27.2% average improvement in evaluation consistency with human-annotated benchmarks (Pearson correlation from 0.655 to 0.833).

**Questions:**

1. Have you compare the annotations of o3-mini with human annotators?
2. How would results change if genuine human raters supplied labels for a subset of samples? A small‑scale human audit would bolster the claim that Silencer improves true benchmark quality.
3. Have you tested Silencer on free‑form generation benchmarks (e.g. writing or coding tasks) where label calibration is non‑trivial?
4. How does Silencer perform on difficult datasets?
4. The ensemble assumes bias correlates with consistency. What happens when one generator is much weaker (or stronger) than all others?

**Ethical Concerns:**

["NO or VERY MINOR ethics concerns only"]

**Final Justification:**

The rebuttal has addressed most of my concerns.

**Limitations:**

The authors acknowledge extra compute cost.

**Quality:**

2

**Strengths And Weaknesses:**

Strengths:
1. This paper explores sources of self-bias in LLM-as-a-judge, which is important but underexplored.
2. The silencer framework combines sample-level prompting strategies with a benchmark-level ensemble algorithm that dynamically estimates and neutralizes bias.
3. The experiment result shows lower bias and better consistency with human-annotated benchmarks.


Weaknesses:
1. My main concern is that the experiments adopt o3-mini as a proxy of human annotators. o3-mini still has huge bias.
2. The experiments are conducted only on three datasets that are not difficult enough. Results may vary on datasets that are super difficult. Besides, these three datasets may not fully capture broader domains (e.g., code generation or multimodal tasks)

---

> ### Author Rebuttal · Authors · 2025-07-30
>
> **Dear Reviewer 8ACx, thank you for recognizing the motivation, method and experimental results of our work. Also, we sincerely appreciate you taking the time to read our manuscript and provide valuable feedback. Below are our responses to all the concerns you raised, which will be incorporated into the updated version to further enhance the quality of our submission.**
>
> ---
> > **`My main concern is that the experiments adopt o3-mini as a proxy of human annotators. o3-mini still has huge bias. Have you compared the annotations of o3-mini with human annotators? How would results change if genuine human raters supplied labels for a subset of samples? A small‑scale human audit would bolster the claim that Silencer improves true benchmark quality.`**
>
> We fully understand your concern. In Section 3.2, when quantifying the Question Domain Sub-bias, due to the large scale of the test data (over 1,000 samples), we employed o3-mini as a human proxy to (1) paraphrase the generated questions and (2) provide labels for these questions. We understand your concern regarding whether using o3-mini might introduce additional bias in terms of language style  and label that could affect the results. Below, we analyze the impact in details:
>
> (1) The questions generated by the LLM generator inherently carry both Question Domain Sub-bias and Language Style Sub-bias. To eliminate the Language Style sub-bias, we used o3-mini to paraphrase these questions. To analyze whether the paraphrased questions may introduce an o3-mini-specific Language Style Bias, we calculate as follows:
>
> $\mathcal{B}^{s}_{\mathcal{M}^{\text{o3-mini}}}(\mathcal{M}) = R(\mathcal{M}|\text{para}\_{\mathcal{M}^{\text{o3-mini}}}(\mathcal{M}^{\text{human}}), \mathcal{M}^{\text{human}},\mathcal{M}^{\text{ref}}\_{1:K}) - R(\mathcal{M}|\mathcal{M}^{\text{human}}, \mathcal{M}^{\text{human}},\mathcal{M}^{\text{ref}}\_{1:K})$
>
> This metric evaluates the performance difference of the seven evaluated LLMs on the o3-mini-paraphrased human benchmark versus the original human benchmark. The results show that for Math Reasoning and Language Understanding tasks, $ \mathcal{B}^{s}\_{\mathcal{M}^{\text{o3-mini}}}(\mathcal{M}) $ is measured as **-0.0009** and **0.0004**, respectively. In comparison, $ \mathcal{B}^{q} $ is **0.0247** and **0.0077**, respectively. **Thus, the bias introduced by o3-mini paraphrasing is negligible and does not affect the experimental conclusions.**
>
> (2) We understand your concern that o3-mini’s labels might still contain errors, potentially with bias, and thereby affect the measured Question Domain sub-bias. Following your suggestion, we conducted a human audit. For each sub-task, we randomly sampled 20 generated questions and manually (at least with a bachelor’s degree) verified o3-mini’s labels. The audit revealed that the error rates of o3-mini’s labels were **7.8%** for Math Reasoning and **6.9%** for Language Understanding. We further tested whether these labeling errors introduced bias using:
>
> $$\mathcal{B}^{l}\_{\mathcal{M}^{\text{o3-mini}}}(\mathcal{M}) = R(\mathcal{M}|\mathcal{M}, \mathcal{M}^{\text{o3-mini}},\mathcal{M}^{\text{ref}}\_{1:K}) - R(\mathcal{M}|\mathcal{M}, \mathcal{M}^{\text{human}},\mathcal{M}^{\text{ref}}\_{1:K})$$
>
> The results show that for Math Reasoning and Language Understanding tasks,
> $ \mathcal{B}^{l}\_{\mathcal{M}^{\text{o3-mini}}}(\mathcal{M}) $ is **0.0014** and **0.0007**, respectively.
> Again, these values are negligible compared to $ \mathcal{B}^{q} $ (**0.0247** and **0.0077**), **indicating that the labeling bias introduced by o3-mini is minimal.**
>
> **In summary,  the experimental evidence demonstrates that using o3-mini as a human proxy for measuring sub-bias does not affect the conclusions presented in Section 3**. This is intuitively reasonable: models with training trajectories more similar to o3-mini may benefit from its proxy setting, while models more dissimilar to o3-mini may be slightly disadvantaged, thereby leading to the two effects offsetting each other.
>
> We sincerely thank you for this insightful suggestion.
>
> ---
> > **`The experiments are conducted only on three datasets that are not difficult enough. Results may vary on datasets that are super difficult. Besides, these three datasets may not fully capture broader domains (e.g., code generation or multimodal tasks).`**
>
> We understand and agree with your suggestion that conducting experiments on more challenging and diverse tasks (e.g., open-domain tasks) would better validate the generalizability of Silencer. We would also like to clarify that prior works on LLM-as-benchmark-generator generally focused on a limited set of closed-domain tasks. For instance, [1] conducted evaluations on the same three tasks as ours, [2] tested on Math Reasoning, Language Understanding, Commonsense Reasoning, and QA tasks, while [3] covered QA and Gender Bias classification tasks.
>
> This is mainly because assessing LLM-as-benchmark-generator approach is costly (requiring a large number of tester inference results), and closed-source tasks are more conducive to unifying the benchmark generation framework.
>
> Nevertheless, we agree that verifying the generalizability of Silencer on a more challenging, free-form generation task such as **competitive coding** would be valuable. To this end, we chose OJBench [4] as high-quality human-annotated benchmark. We modified BenchMaker’s label generation setting to prompt it to generate 10 test cases that thoroughly evaluate the function’s correctness, thereby enabling support for free-form generation tasks. During Label Calibration, Silencer accepts test cases provided by other LLMs as input, and jointly analyzes and compares all candidate test cases to regenerate calibrated 10 test cases. We followed [4] in evaluating the Pass@1 metric, and chose Python as the programming language with PyPy3 interpreter. All other experimental settings are aligned with Section 5.1.
>
> **Results on Competitive Coding Task**
>
> |Methods|$\mathcal{B}$|$r_p$|
> |-|:-:|:-:|
> |Baseline|0.1336|0.7172|
> |w. Attribute Integration|0.1204|0.7346|
> |w. Cross Paraphrase |0.1098|0.7461|
> |w. Label Calibration|0.1130|0.7487|
> |w. Silencer|0.0169|0.8229|
>
> As shown above, our findings on this task are consistent with other tasks:
> - **All modules of Silencer effectively suppress self-bias, ultimately reducing self-bias to near zero. This significantly improves the alignment between the de-biased benchmark and the human-annotated benchmark (from 0.7172 to 0.8229 in $r_p$).**
> - Among the sub-bias mitigation strategies, Label Calibration brings the greatest improvement in $r_p$.
>
> Meanwhile, unlike other tasks, Cross Paraphrase results in a larger reduction of self-bias. Our case study indicates that the longer questions in competitive coding amplify language style bias, making its mitigation through Cross Paraphrase more effective.
>
> Additionally, we found that the average pass@1 of the tested models was only 27%, which is significantly lower than their accuracies on other tasks (all above 35%). This demonstrates that **Silencer maintains strong generalizability even on the more challenging, free-form generation task of competitive coding**. In the future, given sufficient resources, we plan to evaluate Silencer on a wider range of tasks (such as multi-modal and writing tasks) to further explore its generalizability.
>
> ---
> > **`The ensemble assumes bias correlates with consistency. What happens when one generator is much weaker (or stronger) than all others?`**
>
> We consider this suggestion highly insightful, as it helps explore the stability of Silencer under extreme conditions. To this end, we extend the experimental setup of the Math Reasoning task by separately adding an LLM to the original set of 7 generator LLMs:
> - A weaker LLM, Qwen2.5-7B-Instruct (models weaker than this lack sufficient instruction-following capability to support the task).
> - A stronger LLM, OpenAI o4-mini (the strongest API model available to us, which significantly outperforms other generators on mathematical reasoning tasks).
>
> For both extended experiments, we measure the **Pearson correlation coefficient $r_p$ between the reciprocal of evaluation consistency and self-bias**.
>
> **Extended Results on Math Reasoning Task**
>
> |Settings |  Original 7 Generators |  w. a Much Weaker Generator |  w. a Much Stronger Generator |
> |-|:-:|:-:|:-:|
> |$r\_p$|0.861|0.882|0.873|
>
> We observe that the added weaker generator exhibits the largest self-bias and the lowest evaluation consistency with the ensembled benchmark, which aligns with our original hypothesis and leads to an increase in $r_p$.
> Similarly, the added stronger generator exhibits the smallest self-bias and the highest evaluation consistency with the ensembled benchmark, also resulting in an increase in $r_p$.
>
> Moreover, we find that when adding a much weaker generator, directly ensembling benchmarks from different generators with equal weights significantly reduces the consistency with the human benchmark (from 0.9216 to 0.9008). In contrast, by leveraging the assumption that bias is consistent with evaluation consistency, Silencer substantially suppresses the ensemble weight of the weaker generator, mitigating the impact on the consistency with the human benchmark (only a slight drop from 0.9386 to 0.9369).
> When adding a much stronger generator, Silencer also yields additional benefits (from 0.9386 to 0.9451) compared to equal-weight ensembling (from 0.9216 to 0.9242).
>
> **These results empirically confirm that under extreme conditions, Silencer’s ensemble strategy guided by evaluation consistency leads to higher-quality benchmarks**.
>
> **References**
>
> [1] LLM-Powered Benchmark Factory: Reliable, Generic, and Efficient.
>
> [2] Unigen: A unified framework for textual dataset generation using large language models.
>
> [3] Discovering language model behaviors with model-written evaluations.
>
> [4] OJBench: A Competition Level Code Benchmark For Large Language Models.

---

> ### Author Response · Authors · 2025-08-07
>
> Dear Reviewer 8ACx,
>
> We are writing to you with a gentle follow-up as the discussion period is about to close. We completely understand that your time is precious and you are likely very busy. We just wanted to kindly reiterate that if you have any final comments or concerns, we are fully prepared to do whatever is necessary to address them.
>
> Thank you once again for your time and invaluable contribution to our work.
>
> Best regards,
>
> Authors

---

> > ### Comment · Reviewer_8ACx · 2025-08-08
> >
> > Thank you for your responses which have addressed some of my concerns. My remaining concern is still the usage of o3-mini. The dataset the author uses is relatively easy. For much harder datasets, I'm still concerned with high bias.

---

> > > ### Author Response · Authors · 2025-08-08
> > > **Follow-up Discussion**
> > >
> > > Dear Reviewer 8ACx,
> > >
> > > Thank you for your feedback. In our previous rebuttal, we demonstrated on both Math Reasoning and Language Understanding tasks that the bias introduced by using o3-mini as a human proxy to estimate the Question Domain Sub-bias is several orders of magnitude smaller than the Sub-bias itself, and thus can be regarded as negligible.
> > >
> > > We understand that your remaining concern lies in the possibility that, for more challenging tasks, using o3-mini as a human proxy in estimating the Question Domain Sub-bias might introduce additional bias. To verify this, immediately after receiving your reply, we conducted further tests on a sufficiently difficult competitive coding task, using the high-difficulty OJBench as human-annotated benchmark. On this task, the average pass@1 of the tested models was only 27%, underscoring its difficulty. Following the same verification procedure described in our rebuttal, we found that the bias introduced by o3-mini in terms of language style was **0.0008**, and in terms of wrong labels was **-0.0010**. Compared to the Question Domain Sub-bias of **0.0192**, **the bias introduced by o3-mini remains negligible even for such a difficult task**. This aligns with the reasoning we provided in the rebuttal: models with training trajectories more similar to o3-mini may benefit from its proxy setting, while models more dissimilar to o3-mini may be slightly disadvantaged, leading to the two effects offsetting each other.
> > >
> > > **Having verified that the bias introduced by o3-mini is statistically negligible across various tasks—and having explained the underlying reason—we would like to further emphasize that o3-mini is used only for estimating the value of the Question Domain Sub-bias, in order to help us better understand its potential impact. In all our other experiments, including the main experimental results, o3-mini was not involved as a human proxy.**
> > >
> > > Thank you for taking the time to engage in this discussion. We hope our response help address your concern, and we look forward to your further reply.

---

> > > ### Author Response · Authors · 2025-08-09
> > >
> > > Dear Reviewer 8ACx,
> > >
> > > As the discussion deadline approaches, we would like to kindly ask if you have any further concerns or suggestions. We look forward to continuing the discussion with you.
> > >
> > > Best regards,
> > >
> > > Authors

---

> > > > ### Comment · Reviewer_8ACx · 2025-08-09
> > > >
> > > > Thanks for the new experiment, I decided to raise my score to 4.

---

> > > > > ### Author Response · Authors · 2025-08-09
> > > > >
> > > > > Dear Reviewer 8ACx,
> > > > >
> > > > > Thank you for your time and prompt response. We believe that incorporating the outcomes of our discussion into our manuscript will greatly enhance the quality of our work.
> > > > >
> > > > > Best regards,
> > > > >
> > > > > Authors

---

### Comment · Area_Chair_HC76 · 2025-08-03
**Reminder for author-reviewer discussion**

Hi reviewers,

Thanks for reviewing the paper. Could you take a look at authors' response and reply? Thank you.

Yours,

AC

---

### Author Response · Authors · 2025-08-05
**General Comment**

**We sincerely thank all reviewers for dedicating their valuable time to reviewing our manuscript, as well as for recognizing our proposed Silencer framework and providing helpful suggestions.** We are pleased to note that the reviewers generally acknowledge our strengths:

* **Strong motivation \[8ACx, iMmg, wEJW]**: The issue of self-bias in LLM-as-benchmark-generator is important and has not been sufficiently explored. This work thoroughly investigates the problem and proposes Silencer framework to mitigate it.
* **Sound and insightful analysis \[sV2H, wEJW]**: Our work attributes the causes of self-bias and verifies them one by one through credible experiments, providing profound insights for future research.
* **Effectiveness of the proposed Silencer framework \[8ACx, sV2H, iMmg, wEJW]**: The paper conducts extensive experiments covering various models, tasks, and ablation studies. The results demonstrate the effectiveness of the proposed Silencer framework.
* **Broad significance \[iMmg]**: This work has broad implications for improving the reliability of benchmarks generated by LLMs.
* **Good readability \[wEJW, iMmg]**: The paper is well-written, with detailed explanations and clear logic.

---
We also deeply appreciate the reviewers’ detailed suggestions from different perspectives, which we have addressed one by one during the rebuttal phase. These resolutions will be incorporated into the revised version, which we believe will further enhance the quality of our manuscript:

* **o3-mini as a human annotator proxy may introduce additional bias \[8ACx, sV2H]**: Our experiments during the discussion phase confirmed that o3-mini does not introduce significant bias that affect the results we observed in the preliminary phase, and we provided intuitive explanations. Including this analysis in the paper will strengthen the credibility of our self-bias attribution.
* **Testing Silencer’s effectiveness on broader domains \[8ACx]**: We further evaluated Silencer on competitive coding task, and the results validated its effectiveness on more challenging free-form generation task domain.
* **Additional analytical experiments \[8ACx, wEJW]**: Following reviewer 8ACx’s suggestion, we investigated scenarios where the generator set includes a significantly weaker or stronger generator. Silencer adaptively estimates self-bias and combines it with a weighted ensemble algorithm to minimize self-bias and produce higher-quality benchmarks compared to baseline methods. Based on reviewer wEJW’s suggestion, we explored the evolution of convergence steps and self-bias during algorithm iterations, gaining deeper insights into Silencer’s mechanism.
* **Providing stronger theoretical support for self-bias attribution \[wEJW]**: We incorporated findings from the education domain to theoretically and comprehensively justify decomposing self-bias into three sub-biases, which makes the attribution results more convincing.
* **More details \[sV2H, iMmg]**: Based on reviewer sV2H’s suggestion, we included more fine-grained experimental results and additional examples of model-generated content to provide further insights. Following reviewer iMmg’s suggestion, we conducted a qualitative evaluation of the quality and correctness of generated paraphrases, which offers a better understanding of Silencer’s underlying mechanism.

---
We are happy to provide any additional details to help the reviewers better understand and recommend our manuscript, and we look forward to further feedback.

---

### Note · Authors · 2025-08-16

As this review cycle draws to a close, we would like to once again express our heartfelt thanks to all reviewers for the valuable time and thoughtful suggestions devoted to evaluating our work; these comments have helped us further improve the quality of our manuscript. We also appreciate the reviewers’ clear support for our study, which can help bring the self-bias issue in LLM-as-benchmark-generator to the attention of more researchers in the community.

Meanwhile, we are deeply grateful to the AC for your dedication and timely guidance throughout the review process; under your coordination, all reviewers conscientiously and efficiently fulfilled their responsibilities.

Going forward, we will open-source our code framework and data as soon as possible to facilitate further research in this area.

Best regards,

Authors

---

### Decision · Program_Chairs · 2025-09-17

**Decision:**

Accept (poster)

**Comment:**

The paper studies LLM self-bias in benchmark generation and evaluation. It then proposes Silencer to alleviate this phenomenon. Reviewers agree that the method is effective and the analysis is well-conducted. We would also like to remind the author to incorporate reviewers' comments to the final version.